# Simulated Zonal Current Characteristics in the Southeastern Tropical Indian Ocean (SETIO)

Nining Sari Ningsih[1], Sholihati Lathifa Sakina[2], Raden Dwi Susanto[3,2], Farrah Hanifah[1]

[1]Research Group of Oceanography, Faculty of Earth Sciences and Technology, Bandung Institute of Technology, Indonesia
[2]Department of Oceanography, Faculty of Earth Sciences and Technology, Bandung Institute of Technology, Indonesia
[3]Department of Atmospheric & Oceanic Science, University of Maryland, USA

*Correspondence to*: Nining Sari Ningsih (nining@fitb.itb.ac.id)

**Abstract.** So far, detailed dynamics and characteristics of ocean currents in the Southeastern Tropical Indian Ocean (SETIO)
have not been fully explained because duration of observed currents and number of observation points in the region are relatively limited. In this study, zonal current characteristics in the SETIO adjacent to the southern Sumatra-Java coasts have been studied using data over a relatively long period of time (64 years; 1950-2013), which were derived from simulated results of a 1/8° global version of the HYbrid Coordinate Ocean Model (HYCOM). This study has revealed distinctive features of zonal currents in the South Java Current (SJC) region, the Indonesian Throughflow (ITF)/South Equatorial Current (SEC)
region, and the transition zone between the SJC and ITF/SEC regions. Empirical orthogonal function (EOF) analysis is applied to investigate explained variance of the current data, in which the first three EOF modes accounted for almost 75-98% of total variance. The first temporal mode of EOF is then investigated by using ensemble empirical mode decomposition (EEMD) for distinguishing the signals.

The EOF1 mode of zonal current across three meridional sections, namely East Java (EJ), West Java (WJ), and Sumatra (SM),
clearly shows peculiar features of zonal currents between nearshore and offshore regions in the sections. In Sections EJ and WJ and in the offshore area of Section SM, the vertical structure of EOF1 is characterized by one-layer flow. Conversely, in the nearshore and transition regions of Section SM, it is designated by two-layer flow. Furthermore, the EEMD analysis shows that the zonal current variation in the SJC region has peak energies, which are sequentially dominated by semiannual, intraseasonal, and annual timescales, while semiannual and intraseasonal periods with pronounced interannual variations of
current appear consecutively to be dominant modes of variability in the transition zone between the SJC and ITF/SEC regions. In contrast, there exist dominant interannual period with prominent intraseasonal variability of the current in the ITF/SEC region. In response to El Niño–Southern Oscillation (ENSO) and Indian Ocean Dipole (IOD), it is found that ENSO has a strongest influence on the zonal current fluctuations at $C_{EJ}$ (close to the outflow area of the ITF), with the ENSO leading the current by 4 months. Meanwhile, the IOD is most dominant in influencing the current velocity variations at $B_{SM}$ (close to the
center of eastern pole of the IOD), with the current lagging the IOD by 1 month. Moreover, based on the long-term HYCOM simulation, the proportion of contribution of each EEMD mode to the EOF1 is estimated by calculating standard deviation of each EEMD mode relative to the total variance of first principal component (PC1). It revealed that the contributions of each

EEMD mode to the EOF1 at $A_{WJ}$ (nearshore region) and $B_{SM}$ in the order from largest to smallest are intraseasonal, semiannual, annual, interannual, and long-term fluctuations. Impressively, the contribution of long-term variation (19.2 %) at $C_{EJ}$ is larger than the interannual (16.3 %) and annual (14.7 %) variations. Detailed analyses of long-term variation are beyond the scope of this study. Therefore, future studies should be conducted to investigate the existence of profound contribution of long-term variation to the EOF1 at $C_{EJ}$.

## 1 Introduction

Southeastern tropical Indian Ocean (SETIO) plays an important role in ocean and atmosphere dynamics of Indian Ocean. Several features make the SETIO region unique. This is partly due to the presence of the Indonesian Throughflow (ITF) (Gordon, 1986; Wyrtki, 1987; Murray and Arief 1988; and references hereafter), which transfers warm and fresh Pacific waters to the Indian Ocean and contributes to variability of sea surface temperature (SST) in the SETIO, particularly that in the area off Java and Sumatra, which in turn affects the climate system both at regional and global scales (Clark et al., 2003; Saji and Yamagata, 2003). In the SETIO, the complex dynamical circulations exist due to the coexistence of South Java Current (SJC), South Java Undercurrent (SJUC), South Equatorial Current (SEC), and also the ITF originating from the outflow passages (e.g., Sunda, Lombok, and Ombai Straits, and Timor Passage) and their mutual interactions. It has been recognized that the SJC and SJUC play an important role in distributing warm and fresh water into and out of the southeast Indian Ocean and in turn influencing the global climate system (e.g., Fieux et al., 1994, 1996; Sprintall et al., 1999, 2010; Wijffels et al., 2002; Wijffels and Meyers, 2004).

Previous studies have suggested that the current dynamics in the SETIO as well as ocean circulations in the inner Indonesian seas are strongly linked to the regional Indo-Pacific and global climates from intraseasonal, seasonal, interannual, and even longer time scales (e.g., Sprintall et al., 1999; Song et al., 2004; Iskandar et al., 2006; Yuan et al., 2008; Syamsudin and Kaneko, 2013; Sprintall and Révelard, 2014; Krishnamurthy and Krishnamurthy, 2016; Susanto et al., 2016). On intraseasonal time scale, Iskandar et al. (2006) have confirmed the existence of intraseasonal variations of SJC and its deeper undercurrent (SJUC) along the southern Sumatra-Java coasts using simulations from an ocean general circulation model (OGCM) for 13 years (1990–2003). They found that the intraseasonal SJC is dominated by the 90-day variations associated with propagation of the first baroclinic Kelvin waves, which are driven by strong 90-day winds over the central equatorial Indian Ocean. Meanwhile, 60-day variations are the dominant feature in the SJUC, which are forced by intraseasonal atmospheric variability associated with the eastward movement of the Madden-Julian Oscillation (MJO) over the eastern equatorial Indian Ocean.

Meanwhile, seasonal variabilities of SJC and SJUC those exist along the coasts of western Sumatra and southern Java have been investigated based on observation data (e.g., Sprintall et al., 1999; 2010; Qu and Meyers, 2005). In general, their studies have revealed that the SJC is eastward during the northwest (NW) monsoon (December to February; DJF) and the eastward-flowing SJC is enhanced in the presence of semiannual coastal Kelvin waves originating in the equatorial Indian Ocean during the first (March to May; MAM) and second (September to November; SON) transitional monsoons. During the southeast (SE)

monsoon (July to August; JJA), the SJC flows mostly westward. In addition, Sprintall et al., (2010) have confirmed the extension of SJC and SJUC into Ombai Strait through Sawu Sea based on 3-year velocity measurements (2004–2006).

Moreover, like SJC, ITF also has seasonal variability. Sprintall et al. (2009) have examined the ITF transport in three exit passages, namely Lombok and Ombai Straits, and Timor passage using INSTANT (International Nusantara STratification ANd Transport) data from January 2003 through December 2006. Their results show that seasonal variations of the ITF are

influenced by the monsoon climate, with maximum ITF occurs during the SE monsoon. Under El Niño–Southern Oscillation (ENSO) cycle, interannual variability of ENSO also affects the ITF transport, in which ENSO-related wind forcing is found to modulate the variability of ITF transport, which strengthened (weakened) during La Niña (El Niño) (Susanto et al., 2012; Susanto and Song, 2015; Feng et al., 2018). In addition to ENSO, Pujiana, et al. (2019) have revealed that Indian Ocean Dipole (IOD) was also responsible for the anomalous ITF. They found a reduction in the ITF transport in 2016 due to an unprecedented

negative IOD event. Feng et al. (2018) also reported the presence of decadal and interdecadal variations of the ITF transport, which is mostly due to the ITF responses to atmospheric forcing (trade winds) and oceanic adjustment in the Pacific (Meng et al., 2004; Feng et al., 2018). In addition to the wind forcing mechanism, fluctuations in rainfall over the Indonesian Seas that modulates salinity also influences the ITF transport on interannual (Hu and Sprintall, 2016) and decadal (Hu and Sprintall, 2017; Jyoti et al., 2019) time scales. They found that the salinity effect mechanism is an important component of ITF dynamics

and it is different from the wind forcing mechanism. Moreover, it has been revealed that salinity effect contributes 36% of the total interannual variability of the ITF transport (Hu and Sprintall, 2016) and dominates an increasing trend of the ITF transport during the past decade (Hu and Sprintall, 2017).

In the offshore area of the SETIO, it has been reported that SEC in south of Java has intraseasonal variation on a 60-day timescale (e.g., Quadfasel and Cresswell, 1992; Semtner and Chervin, 1992; and Bray et al., 1997). Further research carried

out by Feng and Wijffels (2002) shows that baroclinic instability seems to be the main cause of intraseasonal variability in the SEC. Moreover, it is known that SEC in the southern Indian Ocean bifurcates at the east coast of Madagascar into the Northeast (NEMC) and Southeast Madagascar Currents (SEMC). Yamagami and Tozuka (2015) have investigated interannual variability of the SEC bifurcation along the Madagascar coast. Their results indicate that interannual variation of SEC bifurcation latitude and the NEMC and SEMC transports are correlated with Niño 3.4 index, with a lag of about 5–15 months. However, the

seasonal and interannual variations of SEC in the SETIO are unclear yet.

Regarding dynamics and characteristics of the SETIO, especially adjacent to the western coast of Sumatra and the southern coast of Java, previous studies are either based on numerical model, remote sensed data, or velocity/moorings observations within the Indonesian seas or at the exit passages of Indonesian seas (Sunda, Lombok, Ombai, and Timor passages), which lead into the SETIO. There is almost no ocean current/velocity measurement within the SETIO. The observational velocity

data are available only at limited points in space and time. The first velocity measurement in the SETIO region was reported by Sprintall et al. (1999). The mooring was deployed in 200 m water depth off the south coast of Java at 109.53° E, 8.19° S from March 1997 to March 1998 at depths of 55 m, 115 m, and 175 m, but only current meters at 115 m and 175 m were fully working properly (Sprintall et al., 1999). It should be underlined that the period of velocity measurement was conducted during

strong El Niño/positive IOD episodes. Hence, not only might the observed currents not characterize the neutral years, but its

characteristics might also not be fully resolved due to this limited vertical resolution. Another velocity measurement in the south coast of Java with a relatively higher vertical resolution is collected by RAMA (Research Moored Array for African-Asian-Australian Monsoon Analysis and Prediction). The RAMA mooring was installed at 106.75°E, 8.5°S (indicated by point $R_2$ in Fig. 1) and it provides current data for a period of 17 months (December 2008 to May 2010) from the near surface down to a depth of 136 m with vertical resolution of 8 m. Due to this limited duration of observed currents, it might hard to resolve

variations on time scales greater than the semiannual cycle. Recently, there are some moorings to measure velocity and stratification deployed in the SETIO region. However, they have not been fully recovered nor published. Therefore, due to the limited duration of in situ velocity measurements and the limited number of observation points in the SETIO, the detailed dynamics and characteristics of ocean currents in the region are not fully understood yet. It is important to obtain a better understanding of current characteristics as well as their spatial and temporal variations in the SETIO adjacent to the southern

coasts of Sumatra and Java both for scientific and practical reasons, such as fisheries, climate, and navigation. These are the main motivations of the present study.

In addition, many studies of the current dynamic in the SETIO adjacent to the southern coasts of Sumatra and Java, which were carried out by the previous investigators mentioned above (i.e., Sprintall et al., 1999, 2010; Qu and Meyers, 2005; Iskandar et al., 2006), focused on intraseasonal and seasonal variations based on relatively limited observation periods and

measured data points. To the best of our knowledge, researches concerning features of zonal currents in the SETIO, especially in regions of SJC, ITF/SEC, and transition zone between the SJC and ITF/SEC as well as their interannual and long-term variations, have so far not been extensively studied in the regions, both based on observations and numerical models. It is necessary to acquire better and comprehensive insights of both spatial and temporal characteristics of the current circulation in the region. Hence, the aims of this paper are: (1) to further investigate basic features and mode structures of the current

vertical profile time series and their temporal variability in the SETIO adjacent to the Sumatra-Java southern coasts based on relatively long-term data (64 years) derived from simulated results of a 1/8° global version of the HYbrid Coordinate Ocean Model (HYCOM), (2) to better understand variability of the zonal current in the area of study, especially on intraseasonal, seasonal, and interannual timescales by using a combination of the Empirical Orthogonal Function (EOF) analysis and the Ensemble Empirical Mode Decomposition (EEMD) method (i.e., Huang et al, 1998; Wu and Huang, 2009; Shen et al., 2017,

and references thereafter), whereas its long-term variation is beyond the scope of this study, and (3) to discuss exclusively the ocean current characteristics in the SETIO and subsequently elaborate their genesis.

## 2 Data and Methods

The HYCOM has been successfully used by previous investigators to simulate current circulation within the Indonesian waters (e.g., Gordon et al., 2008; Metzger et al., 2010; Shinoda et al., 2012). In this study, we analyzed the monthly mean HYCOM

simulated currents with 1/8° horizontal resolution for the period of 64 years (1950-2013). Simulation results of the HYCOM

version used in this study have been verified against several data and the verifications have been documented in our earlier publications (Hanifah and Ningsih, 2016). In addition to the aforementioned comparisons, in this paper we have performed comparisons between the moored RAMA provided by the NOAA and HYCOM currents at two points (marked by points $R_1$ and $R_2$), and also comparisons between OSCAR (Ocean Surface Current Analysis Real-time) and the HYCOM currents at three points (marked by points $O_1$, $O_2$, and $O_3$), as shown in Fig. 1. The RAMA and OSCAR datasets have been provided by the NOAA (https://www.pmel.noaa.gov/tao/ data_deliv/deliv-nojava-rama.html) and Physical Oceanography Distributed Active Archive Center (PODAAC) (https://podaac.jpl.nasa.gov/dataset/OSCAR_L4_OC_third-deg), respectively. The general agreement between the HYCOM currents and those of the moored RAMA is reasonably encouraging with correlation coefficient ($r$) ranging from 0.40 to 0.57 at point $R_1$ (Figs. 1e-h) and 0.49 to 0.55 at point $R_2$ (Figs. 1i-k), with the 95% significance level at both points approximately ±0.04 and ±0.09, respectively. In addition, the root mean square errors (RMSE) between them range from 0.10 to 0.28 m s$^{-1}$ at point $R_1$ and 0.17 to 0.29 m s$^{-1}$ at point $R_2$. Meanwhile, the comparisons between the HYCOM currents and the OSCAR data show general agreement as well at points $O_1$ ($r$=0.65; RMSE=0.17 m s$^{-1}$), $O_2$ ($r$=0.59; RMSE=0.19 m s$^{-1}$), and $O_3$ ($r$=0.60; RMSE=0.21 m s$^{-1}$), with the 95% significance level at the three points ±0.13 (Figs. 1b-d). Further details of numerical model description of this applied HYCOM version can be found in Hanifah and Ningsih (2016). In addition to the HYCOM simulated currents, to support analysis in this research, the Oceanic Niño and Dipole Mode Indices (ONI and DMI, respectively) were used to identify climate conditions and influences of interannual forcing associated with ENSO and IOD on interannual variability of the zonal currents in the study region. The ONI and DMI were obtained from the National Oceanic and Atmospheric Administration (NOAA) website (http://www.cpc.ncep.noaa.gov/data/indices/) and the Japan Agency for Marine Earth Science and Technology (JAMSTEC) website (http://www.jamstec.go.jp/frcgc/research/d1/iod/iod/dipole_mode_index.html), respectively. In addition, the wind fields derived from the NOAA (https://www.esrl.noaa.gov/psd/data/gridded/data.ncep.reanalysis.derived.surface.html) are also used to investigate the effects of local and remote winds on zonal current variations.

The EOF method (i.e., Kantha and Clayson, 2000; Hannachi, 2004) was then used to investigate the mode structure of the zonal current vertical profile and its temporal variability, particularly at points $A_{SM}$, $A_{WJ}$, and $A_{EJ}$ (Transect A), points $B_{SM}$, $B_{WJ}$, and $B_{EJ}$ (Transect B), and points $C_{SM}$, $C_{WJ}$, and $C_{EJ}$ (Transect C), as shown in Fig. 2. Moreover, temporal variability of the first EOF mode of zonal current was analyzed by applying the EEMD method for decomposing a signal into a series of intrinsic mode functions and investigating the zonal current variability in the SETIO region adjacent to the southern coasts of Sumatra-Java. Furthermore, a power spectral analysis (Emery and Thomson, 2001) was applied to the EEMD results to identify dominant periods of the zonal current variability in the study area. The power spectral analysis is computed from a measured time series by cutting the time series into several segments and applying Fourier analysis to these segments. The contribution from individual Fourier harmonics was subsequently summed to derive total energy of time series. In addition, 95% confidence red noise level in power spectrum, specified to acquire accurate confidence thresholds for true periodic signatures, was calculated based on number of degrees of freedom in each frequency band (Mann and Lees, 1996).

## 3 Results

### 3.1 Distinctive Features of Zonal Currents in the Study Area

As we are interested in investigating characteristics of main ocean currents those exist in the SETIO adjacent to the Sumatra-Java southern coasts, such as SJC, ITF, and SEC, in this study we only considered major component of those currents, namely the zonal current component in which it was analyzed from surface to 800 m. The maximum depth of 800 m was chosen to capture the presence of prevailing ocean currents in the area of study and the surrounding regions, such as cores of the SJUC. For example, these cores in the Ombai Strait exist at about 400–800 m depth (Sprintall et al., 2010). Furthermore, based on monthly averaged surface currents over 64 years period (1950–2013), we analyzed the zonal currents at three transects, namely Transects A, B, and C, which represent coastal region, transition zone between coastal and offshore regions, and the offshore one, respectively (Fig. 2). Transects A and C were selected with respect to the prevalence of ocean currents in the area of interest, representing nearshore (SJC) and offshore (ITF/SEC) areas, respectively (Qu and Meyers, 2005; Fang et al., 2009; Ding et al., 2013). In the present study, we have performed additional analyses of current characteristics on Transect B as the transition zone between the SJC region (Transect A) and ITF/SEC region (Transect C) due to the existence of typical features of zonal currents along the three transects (A, B, and C), as shown in Fig. 2.

To support our reasons for assigning the three transects, we have provided Fig. 3 (as an example), which clearly shows the particular features of near-surface zonal currents along the three transects. Dynamics of zonal surface currents on Transect A (Fig. 3c), especially along the southern coasts of Sumatra-Java (98° E–114° E), show a complex interplay between remote wind forcings from both the equatorial Indian and Pacific Oceans and local wind. In general, there exist enhanced eastward-flowing currents during MAM and SON, which are probably attributed to Kelvin wave passage. Seasonal characteristic of zonal currents associated with local wind, which is eastward (westward) during DJF (JJA), especially along the southern coast of Java, can be clearly seen after 6–12 months band-pass filtering (figure not shown). In contrast, westward currents are dominant along Transect C (Fig. 3e). Meanwhile, although westward currents are quite dominant along Transect B, eastward currents are also present, especially at longitudes 95° E to 107° E (Fig. 3d). Here, longitude-depth plots of mean zonal currents along the sections A, B, and C are also presented in Fig. 4, which clearly shows different zonal current system along the transects. Mean zonal currents along Transect A (Fig. 4a) show two distinguishing features: (1) the mean currents dominantly flow eastward from the sea surface to 100 m depth (95° E–114° E), and (2) they are predominantly westward from the region (115° E), which is close to Lombok Strait (LS) as one of the ITF exit passages, to the 122° E longitude line. In addition, the mean eastward current at $A_{EJ}$ also exists at depth beneath 100 m and reaches of about 0.03 m s$^{-1}$ at ~400 m. Meanwhile, the average current on Transect B (the transitional zone) is westward, especially at longitudes 101° E to 107° E (Fig. 4b). In the offshore region (Transect C), mean zonal current flows westward throughout the region (Fig. 4c).

Moreover, we also presented meridional sections of zonal current along the three longitudes (yellow lines in Fig. 2) to justify the selection of the locations for analyzing zonal current characteristics, namely sections Sumatra (SM; 98° E); West Java (WJ; 107° E); and East Java (EJ; 113° E), as shown in Fig. 5 (as an example). Figure 5 clearly shows the typical features of near-

surface zonal currents along the three meridional sections, namely the coastal (SJC) area ($0°$ S – ~$2.5°$ S at SM; ~$7°$ S – $8.5°$ S at WJ; and ~$8°$ S – $9.5°$ S at EJ), the transitional zone (~$2.5°$ S – $9°$ S at SM; ~$8.5°$ S –$10°$ S at WJ; and ~$9.5°$ S – $10.5°$ S at EJ); and the offshore (ITF/SEC) area (~$9°$ S – $12°$ S at SM; ~$10°$ S –$12°$ S at WJ; and ~$10.5°$ S – $12°$ S at EJ).

Furthermore, because we are specifically interested in zonal current characteristics off southern waters of Sumatra and Java, we selected three points on each transect, namely points $A_{SM}$, $A_{WJ}$, and $A_{EJ}$ on the Transect A; points $B_{SM}$, $B_{WJ}$, and $B_{EJ}$ on the Transect B; and points $C_{SM}$, $C_{WJ}$, and $C_{EJ}$ on the Transect C with respect to the particular features of zonal currents shown in Fig. 2, Figs. 3c-e, and Figs. 4-5. Here, the subscripts SM, WJ, and EJ of the nine selected points represent regions, which close to Sumatra, West Java, and East Java, respectively.

**3.2 Climatological Current Fields**

Based on the unique features of near-surface zonal currents along the three meridional sections (EJ: $A_{EJ}$-$B_{EJ}$-$C_{EJ}$; WJ: $A_{WJ}$-$B_{WJ}$-$C_{WJ}$; and SM: $A_{SM}$-$B_{SM}$-$C_{SM}$ in Fig. 2) as shown in Fig. 5, we further investigated vertical structure of zonal current along the sections. Figure 6 shows seasonal mean profiles of zonal current velocity and its average (the climatological current field) over the period of 64 years (1950–2013). Seasonal variations of the zonal currents were analyzed during DJF, MAM, JJA, and

SON at each point (Sections EJ, WJ, and SM), as shown in Fig. 2. It can be clearly seen in Fig. 6 that there are special characteristics of the mean zonal currents on each meridional transect (denoted by black lines in the Fig. 6). In the following subsections, we analyzed the climatological current fields of each meridional transect.

**3.2.1 Vertical Structure of Zonal Current along Meridional Section EJ ($A_{EJ}$-$B_{EJ}$-$C_{EJ}$)**

Different zonal current system along the meridional transect East Java (EJ; $A_{EJ}$-$B_{EJ}$-$C_{EJ}$) can clearly be seen in Figs. 6a-f. On

average, for the period 1950 through 2013, zonal climatological current at $A_{EJ}$ (nearshore area) generally flows eastward from the sea surface to 100 m depth (Figs. 6a and 6d) and reaches its maximum value of about 0.16 m s$^{-1}$. It is suggested that the average zonal current at this point is mainly attributed to SJC and it shows seasonal variations. During the SE monsoon (JJA), the strength of climatological eastward SJC at this point in upper 10 m depth reduces (Fig. 6d). Meanwhile, during the NW monsoon (DJF), the current in the upper 10 m (Fig. 6d) flows more eastward in response to the prevailing northwesterly winds

(Fig. 7). In general, the mean eastward current at $A_{EJ}$, during DJF was attributed to local winds. Interestingly, during this monsoon period (DJF), the eastward current at $A_{EJ}$, particularly that at depth beneath 100 m, strengthens and occurs up to ~800 m. Other physical processes may account for the enhanced eastward current at this point. The SJC and SJUC, which are seasonally varying currents and predominantly eastward, are defined as the surface current in the upper 150 m and the subsurface current beneath 150 m down to 1000 m, respectively (Iskandar et al., 2006). The eastward-flowing SJC and SJUC

are intensified, coinciding with the arrival of a seasonal downwelling Kelvin wave along the south coast of Java (e.g., Sprintall et al., 1999, 2000; Iskandar et al., 2006). Downwelling Kelvin waves originating in the equatorial Indian Ocean during the transitional monsoons propagate along the coasts of western Sumatra and southern Java with phase speeds ranging from 1.5 to 2.9 m s$^{-1}$ (e.g., Sprintall et al., 2000; Syamsudin et al., 2004; Iskandar et al., 2005). These phase speeds indicate that the

downwelling Kelvin waves will arrive at $A_{EJ}$ in 21 – 41 days. In this case, downwelling Kelvin waves generated during the monsoon transition period in November may arrive at $A_{EJ}$ in December/January. Therefore, in addition to the local eastward winds, the downwelling Kelvin waves may also contribute to strengthen the eastward currents at $A_{EJ}$ during the NW monsoon, including those at depth beneath 100 m.

Meanwhile, the average current at $B_{EJ}$ (the transitional zone) is westward. It is suggested that the mean westward current at the point $B_{EJ}$ is more dominated by the ITF (shown by black lines in Figs. 6b and 6e). Based on observation in the exit passages (Lombok Strait, Timor Passage, and total ITF along exit passages), ITF in JJA is stronger than that in DJF (e.g., Sprintall et al., 2009). In this study, however, it is found that westward current at the point $B_{EJ}$ at 100 m depth is stronger during DJF than JJA. This phase changing (delay) of the ITF seasonality from JJA to DJF at this point is also found in the Ombai Strait as documented by Sprintall et al. (2009, their Table 3; 2010, their Fig. 3). Moreover, Sprintall et al. (2010) found cores of subsurface maximum ITF during DJF extending from 100–250 m (100–800 m) depth at the northern (southern) part of the strait. In this present study, this seasonal feature of the subsurface maximum ITF is also found at $B_{EJ}$ in which the corresponding westward current at this point reaches its maximum values at ~100 m depth and the maximum westward current is stronger during DJF than JJA (Figs. 6b and 6e). Hence, we suggest that the primary driver for zonal westward current at $B_{EJ}$ is the ITF coming from the southern Ombai Strait. To confirm the above relation, we have calculated the correlation between zonal westward current at a depth of ~100 m at point $B_{EJ}$ and that representing subsurface (~200 m) maximum ITF in the southern Ombai Strait (Sprintall et al., 2010). The correlation coefficient between the zonal westward current at ~100 m at the $B_{EJ}$ and that of the southern Ombai Strait is 0.58 with the 95% significance level approximately ±0.33. This study shows that the zonal westward current at 100 m depth at $B_{EJ}$ has a strong correlation with the subsurface (~200 m) maximum ITF in the southern Ombai Strait, confirming that the ITF flowing from the Ombai Strait is the primary driver for zonal westward current at $B_{EJ}$.

In the offshore region of the study area, zonal current at $C_{EJ}$ (Figs. 6c and 6f) flows westward throughout the year and has average velocity around 0.20 m s$^{-1}$ in the upper 100 m. Under such characteristics, we supposed that the westward current at this point is the SEC in the southeast Indian Ocean, which joins the ITF flowing out from the Lombok and Ombai Straits, and Timor Passage. The HYCOM westward current at this point is stronger during JJA than DJF, which is associated with seasonal characteristics of the ITF in Lombok Strait, Timor Passage, and of the total ITF through the Lombok and Ombai Straits, and Timor Passage (Potemra, 1999; Sprintall et al., 2009). The westward current at $C_{EJ}$ (Figs. 6c and 6f) reaches its maximum value of about 0.31 m s$^{-1}$.

### 3.2.2 Vertical Structure of Zonal Current along Meridional Section West Java ($A_{WJ}$-$B_{WJ}$-$C_{WJ}$)

Figures 6g-l show vertical structure of zonal current along the meridional transect West Java (WJ; $A_{WJ}$-$B_{WJ}$-$C_{WJ}$). Similar to $A_{EJ}$, mean zonal current at $A_{WJ}$ (nearshore region) is attributed to SJC, which generally flows eastward in upper 100 m depth (Figs. 6g and 6j) and reaches its maximum value of about 0.12 m s$^{-1}$. Our simulation shows that during the monsoon transitions (MAM and SON), SJC is eastward and intensified by the propagation of coastal Kelvin waves associated with the Wyrtki Jet in the equatorial Indian Ocean, which is forced by the local equatorial zonal winds during both monsoons. These waves

propagate along the Sumatra-Java coast (i.e., Sprintall et al., 2000; Druskha et al. 2010, Iskandar et al. 2009) and some portions propagate northward into the Lombok and Makassar Straits (Susanto et al., 2000; 2012; Pujiana et al., 2013), whereas the remaining parts continue eastward (Syamsuddin et al., 2004). Furthermore, the present study shows that the eastward current during SON is stronger than that during MAM, which is consistent with mooring observation in the Makassar Strait (Susanto et al., 2012; their Fig. 3). The stronger eastward current during SON was supposed to be attributed to the faster and more intense climatological Wyrtki Jet during SON than that during MAM (Knox, 1976; McPhaden, 1982; Han et al., 1999; Qiu et al., 2009; McPhaden et al., 2015; Figs. 1d and 2e of Duan et al., 2016) and also associated with the stronger wind forcing over the eastern equatorial Indian Ocean during the SON compared with the MAM period (figure not shown), which is responsible for the Jet.

Moreover, it can be seen that during the NW monsoon the eastward current at $A_{WJ}$ (Figs. 6g and 6j) is weaker than that at $A_{EJ}$ (Figs. 6a and 6d). The weaker current at $A_{WJ}$ may exist as a consequence of the weaker mean NW monsoon at this point compared with that at $A_{EJ}$ (Fig. 7). Interestingly, at a depth of 100 m, there is a maximum westward current at $A_{WJ}$ during DJF with velocity of about 0.1 m s$^{-1}$ (Figs. 6g and 6j). Here, we suggest that ITF is the cause of the westward current at 100 m at $A_{WJ}$ during the DJF. In regard to the ITF, Fig. 3 of Sprintall et al. (2010) shows cores of subsurface maximum ITF extending from 100 m to 250 m depth in the northern part of the Ombai Strait and from 100 m to 800 m depth at the southern part of the strait during DJF. Meanwhile, the influence of ITF on the zonal current at $A_{EJ}$ at 100 m is weaker as a consequence of the stronger NW monsoon at $A_{EJ}$ compared with those at $A_{WJ}$ (Fig. 7), so that the current flows rather eastward at $A_{EJ}$ during DJF (Figs. 6a and 6d).

To further investigate which one is more influential between the ITF and the NW monsoon to force the zonal current at the $A_{WJ}$ and $A_{EJ}$ at 100 m depth, we have carried out correlation between the zonal current at both points (each at depth of ~100 m) and each the NW zonal wind and the zonal current representing subsurface (~200 m) maximum ITF in the southern Ombai Strait (Table 1). Here, the ITF in the southern part of the Ombai Strait was chosen for carrying out the correlation because the ITF flows mainly through the southern part of the passage (Sprintall et al., 2010). It was observed that the subsurface maximum ITF during DJF exists at a depth of about 200 m in both the northern and southern parts of the Ombai Strait and it is stronger during DJF than JJA in both parts of the strait (Fig. 3 of Sprintall et al., 2010). In this study, the DJF zonal currents in the period of 2004 through 2006 in the southern Ombai Strait derived from the INSTANT program (http://www.marine.csiro.au/~cow074/ instantdata.htm) were used for the correlation analysis.

It is found that during DJF the zonal current at $A_{WJ}$ at 100 m shows high correlation with the subsurface (~200 m) maximum ITF in the southern Ombai Strait, whereas its correlation with the NW zonal wind is weak (Table 1). Moreover, although during DJF the correlations between the zonal current at $A_{EJ}$ at 100 m and each the NW zonal wind and the subsurface (~200 m) maximum ITF in the southern Ombai Strait are below the significance level, the NW zonal wind is more influential to force variation of zonal current at $A_{EJ}$ at 100 m than the ITF. Hence, during DJF we suggest that the westward current simulated at $A_{WJ}$ at 100 m is ITF-related, whereas that at $A_{EJ}$ is relatively NW zonal wind-related. As already discussed, in addition to the

295 local eastward winds during DJF, it is suggested that the arrival of downwelling Kelvin waves in December/January at $A_{EJ}$ may contribute to a net eastward current across the water column, which in turn reducing the influence of ITF at this point.

In the transition region, the mean current at $B_{WJ}$ is westward and it is more dominated by the ITF (denoted by black lines in Figs. 6h and 6k). Similar to $B_{EJ}$, the seasonal feature of the subsurface maximum ITF is also found at $B_{WJ}$ in which the corresponding westward current at this point reaches its maximum value at ~100 m depth and it is stronger during DJF than

300 JJA (Figs. 6h and 6k). In this study, it is also found that the zonal westward current at 100 m depth at $B_{WJ}$ has a strong correlation with the subsurface (~200 m) maximum ITF in the southern Ombai Strait, with correlation coefficient about 0.77 and with the 95% significance level approximately ±0.33, corroborating that the ITF flowing from the Ombai Strait is the main driver for zonal westward current at this point.

Furthermore, like $C_{EJ}$, characteristic of persistent westward current exists in the offshore region ($C_{WJ}$), which is attributed to

305 SEC and the westward current has mean velocity around 0.22–0.33 m s$^{-1}$ in the upper 100 m (Figs. 6i and 6l). The simulated westward current at $C_{WJ}$ shows seasonal variations and reaches its maximum value of about 0.48 m s$^{-1}$.

### 3.2.3 Vertical Structure of Zonal Current along Meridional Section Sumatra ($A_{SM}$-$B_{SM}$-$C_{SM}$)

Vertical structures of zonal current along the meridional transect Sumatra (SM; $A_{SM}$-$B_{SM}$-$C_{SM}$) are shown in Figs. 6m-r. Similar to $A_{EJ}$ and $A_{WJ}$, mean zonal current at $A_{SM}$ (nearshore region) is eastward, attributed to SJC, and associated with the Kelvin

wave propagation. However, due to $A_{SM}$ located in front of west of Sumatra Island (Fig. 2), which is oriented in the northwest-southeast direction, the meridional component of velocity at this point is also dominant (Figs. 1a and 2). Therefore, zonal currents at $A_{SM}$ are relatively weaker than those at $A_{WJ}$ and $A_{EJ}$, which are located in front of south of Java Island oriented in the west-east direction. For example, during SON, the eastward current reaches its maximum velocity of about 0.05 m s$^{-1}$ at $A_{SM}$ (cyan lines in Figs. 6m and 6p), whereas it is about 0.23 m s$^{-1}$ (at $A_{WJ}$; Figs. 6g and 6j) and 0.20 m s$^{-1}$ (at $A_{EJ}$; Figs. 6a and

6d) at ~30-50 m depths.

Furthermore, results of this study show that a maximum value of the eastward current at $A_{SM}$, $A_{WJ}$, and $A_{EJ}$ is found at a certain depth (at ~30-50 m depths) and this strengthening of eastward flows is supposed to be attributed to a baroclinic Kelvin wave. The baroclinic Kelvin wave propagating vertically and horizontally along its waveguide can exert energy the most at a certain depth (Drushka et al., 2010; Pujiana et al., 2013; Iskandar et al., 2014). According to laboratory experiment observation

conducted by Codiga et al. (1999) and Hallock et al. (2009), Kelvin wave can be trapped in a slope and propagates along an isobath. This phenomenon is known as slope-trapped baroclinic Kelvin wave. Moreover, Kelvin wave which propagates along continental slope with strong stratification can cause strong current velocity. Codiga et al. (1999) also found that this slope Kelvin wave is formed after encountering a canyon-like bathymetry. Meanwhile, Pujiana et al. (2013) shows that Kelvin wave propagation from Lombok Strait to Makassar Strait, across Sunda continental slope, is along isobaths at depths greater than 50

325 m. In this present study, eastward current along the Transect A has maximum current velocity at depth ~30–50 m. Therefore, it is suggested that this maximum eastward current at ~30–50 m depth associated with slope-trapped Kelvin wave, which propagates at that depth along the southern coasts of Sumatra and Java.

In the transition region, characteristic of average zonal current (the climatological current field) at $B_{SM}$ (Figs. 6n and 6q) is different from that at $B_{WJ}$ (Figs. 6h and 6k) and $B_{EJ}$ (Figs. 6b and 6e). The average current at $B_{SM}$ is eastward, while at points $B_{WJ}$ and $B_{EJ}$ it is westward. During NW and transitional periods of the monsoon, zonal current at $B_{SM}$ flows eastward and reaches its maximum velocity of about 0.12 m s$^{-1}$ at a depth of 40 m within the period of SON (Fig. 6q). Meanwhile, during SE monsoon, the zonal current at this point flows westward. In contrast to the mean zonal currents in the nearshore region ($A_{SM}$), it seems that the average zonal current field at $B_{SM}$ is not attributed to SJC. The reason is the $B_{SM}$ location, which is far from the coasts of Mentawai Islands and Enggano Island off the western coast of Sumatra by 430 km. This distance is more than Rossby radius of deformation at this latitude (~90 km). Thereby, Kelvin waves, which affect the SJC variations, do not exist at this point. We suggest that the current variability at $B_{SM}$ is influenced by tropical current systems in the Indian Ocean, such as the Equatorial Counter Current (ECC), Southwest Monsoon Current (SWMC), and Wyrtki Jet. Here, we displayed seasonal averaged surface currents over 64 years (1950–2013) and schematics of the tropical current systems in the Indian Ocean as supporting evidence (Fig. 8).

Figure 8 shows that $B_{SM}$ is located at an area, which is affected by the ECC, SWMC, and Wyrtki Jet. It can be seen in the Fig. 8a that during DJF, surface currents along the equatorial Indian Ocean are dominated by the westward North Equatorial Current (NEC) and the eastward ECC. Meanwhile, during JJA (Fig. 8c), the NEC disappears and the ECC becomes absorbed into the SWMC, which dominantly flows eastward in the northern Indian Ocean (Tomczak and Godfrey, 1994). In addition, during the transitional periods (MAM and SON), the Jet is generated and it causes a strengthening of eastward flows along the equatorial Indian Ocean (Figs. 8b and 8d). This explains the cause of climatological current at $B_{SM}$ flows eastward and reaches its maximum velocity during SON and MAM. These currents (the ECC, SWMC, and Wyrtki Jet) flow eastward before they turn and some part of their flow feed into the SEC in the southern Indian Ocean.

Current characteristics in the offshore region ($C_{SM}$) generally show similarities with those at $C_{WJ}$ and $C_{EJ}$, as shown in Figs. 6o and 6r. The current at $C_{SM}$ is attributed to SEC and flows westward all year round, with mean velocity around 0.18–0.3 m s$^{-1}$ in the upper 100 m. In addition, the strength of westward current at $C_{SM}$ varies seasonally and reaches its maximum value of about 0.42 m s$^{-1}$ during SON (Fig. 6r).

**3.3 Zonal Current Variability**

EOF analysis gives vertical mode structures (spatial mode) and their normalized temporal mode variabilities relative to the mean which influence zonal current variability in the study area. Before performing the EOF analysis, the average value of the current data has been removed (solid black lines in the Figs. 6a-r). To further analyze the zonal current characteristics in the nearshore and offshore areas, and the transition region between them, we examined the EOF modes of zonal current across the three meridional sections (EJ, WJ, and SM). In this paper, we only considered the first mode of EOF (EOF1) analysis since it is associated with the largest percent of the variance. Figure 9 shows vertical structures and their associated temporal variability of EOF1 of zonal currents along the meridional sections. Here, as an example, the temporal variability is only shown for the

last eight-year period of the EOF1 (2006 to 2013). It can be clearly seen that remarkable features of zonal currents are revealed between nearshore and offshore areas in the three meridional sections (Fig. 9).

In general, temporal mode of EOF1 of zonal currents across each meridional section shows intraseasonal and semiannual variabilities both in the nearshore and transition regions, whereas annual and interannual variations exist in the offshore area. However, the vertical structures of EOF1 in each section are quite different. In the nearshore area of Section EJ (Fig. 9a), the

vertical structure of EOF1 is characterized by one-layer flow with a gradual decrease in speed from the surface to 800 m depth, whereas in the transition and the offshore regions the flow velocities decrease more rapidly with depth until they become nearly zero at depths of about 500 m and 300 m, respectively. Meanwhile, in Section WJ (Fig. 9c), the vertical structure of EOF1 is also characterized by one-layer flow in which its unidirectional vertical structure gradually decreases from the surface to a depth of about 450 m in all areas. In contrast, a different vertical structure of EOF1 appears in Section SM (Fig. 9e). In this

section, the vertical structure is characterized by two-layer flow in the nearshore and transition regions with the changeover between the two types of flow occurring at a depth of about 100 m and 200 m, respectively. In addition, in the offshore area of Section SM, the vertical structure of EOF1 displays a unidirectional flow from the surface to a depth of ~500 m.

To examine the EOF modes of zonal currents in more detail, further analysis was performed at three points on each meridional transect, namely points $A_{EJ}$, $B_{EJ}$, and $C_{EJ}$ (Transect EJ); points $A_{WJ}$, $B_{WJ}$, and $C_{WJ}$ (Transect WJ); and points $A_{SM}$, $B_{SM}$, and $C_{SM}$

(Transect SM). Table 2 displays dominant variances at those points. From the Table 2, the first three modes at each point (except $A_{SM}$) already represent ≥ 95% of the total variance. In fact, the first two modes at each point (except $A_{SM}$ and $A_{EJ}$) already represent ≥ 91% of the total variance.

In here, we only consider the first modes of EOF analysis for further analysis since their percent variances (except at point $A_{SM}$) are more than 50% of the total variance (Table 2). Since the temporal variability of the EOF1 contains more than one

frequency (Figs. 9b, 9d, and 9f) and to find out what frequencies are dominant in the EOF1, it was then analyzed by using the EEMD method to decompose the signal. In this study, the EEMD analyses of currents are only presented at one point on each meridional transect, namely $A_{WJ}$ (Transect WJ), $B_{SM}$ (Transect SM), and $C_{EJ}$ (Transect EJ). The $A_{WJ}$, $B_{SM}$, and $C_{EJ}$ points were chosen to investigate SJC variability, interannual variability in the open SETIO, and SEC and ITF variabilities, respectively.

The EEMD analysis of the first temporal EOF mode provides 10 modes/signals in which the first signal of the EEMD result

is the summation of the second to tenth signals, which is the same as the original EOF first temporal mode of zonal currents. Meanwhile, the second–sixth signals of the EEMD result vary from intraseasonal to interannual variabilities. The remaining signals of EEMD result show the long-term variation and trend. Moreover, the proportion of contribution of each EEMD mode to the EOF1 is estimated by calculating standard deviation of each EEMD mode relative to the total variance of PC1 (Figs. 10-12). In general, the contributions of each EEMD mode to the EOF1 at $A_{WJ}$ and $B_{SM}$, from largest to smallest, are intraseasonal,

semiannual, annual, interannual, and long-term (Figs. 10 and 11). Intriguingly, however, the contribution of long-term signal (19.2 %) at $C_{EJ}$ is larger than the interannual (16.3%) and annual (14.7%) signals (Fig. 12). For the scope of this paper, we only focused on the analysis of the EOF1 of zonal current from intraseasonal to interannual timescales. The interesting results

concerning the existence of pronounced contribution of long-term variation to the EOF1 at $C_{EJ}$ will be investigated in a future study.

### 3.3.1 Intraseasonal, Semiannual, and Annual Variations

Figures 10a–b show vertical structure and temporal variability of the EOF1 (58% of total variance) at $A_{WJ}$, respectively. In order to see more clearly temporal variation of the EOF1 in Fig. 10b, we have provided the last eight-year period of the EOF first temporal mode (Fig. 10c, as an example). Current velocity variability relative to the mean flow can be obtained by multiplying the vertical mode structure (Fig. 10a) with the temporal variability (Fig. 10b).

Intraseasonal, semiannual, and annual variabilities of the EOF first temporal mode at $A_{WJ}$ as results of the EEMD analysis are displayed in Figs. 10d-f, where their power spectra (left) show maximum energy at 3-month, 6-month, and 12-month periods, respectively. At this point, the highest power spectrum occurs at semiannual variability (Fig. 10e). In this figure (right), the semiannual variability of the EOF first temporal mode at $A_{WJ}$ clearly shows the presence of an eastward anomaly of the zonal current during the MAM and SON, which may be enhanced by downwelling Kelvin waves associated with the Wyrtki Jet in the equatorial Indian Ocean. Meanwhile, the anomaly of the zonal current at $A_{WJ}$ is westward during JJA in response to the prevailing southeasterly local winds during the SE monsoon. On the other hand, during DJF, the anomaly of the zonal current at $A_{WJ}$ is not associated with the prevailing northwesterly local winds during the NW monsoon, in which the current anomaly is westward during this monsoon (Fig. 10e). As already discussed in Sect. 3.2 (Table 1 and Fig. 7), this may be attributed to the ITF that has more influence on variation of zonal current at $A_{WJ}$ during DJF than the NW local wind.

Similar to $A_{WJ}$, the first mode of EOF vertical structure and its temporal variability (64% of total variance) at $B_{SM}$ show seasonal pattern (Figs. 11a-c). It is also found that signal on a 6-month (semiannual) period is quite dominant at $B_{SM}$ (Fig. 11e). In order to see more clearly the seasonal variation, we have provided a probability distribution function of the EOF1 of zonal currents for each of the NW, SE, and transition seasons at $B_{SM}$ at a depth of ~40 m (Fig. 14). The 40 m depth was selected as an example because the most obvious seasonal variation of currents presents at this depth. It is found that variation of zonal current at $B_{SM}$ is dominantly eastward during DJF (Fig. 14a) and this eastward current is enhanced during MAM and SON (Figs. 14b and d), which may be attributed to the tropical current systems in the Indian Ocean (ECC, SWMC, and Wyrtki Jet). Meanwhile, during JJA (Fig. 14c), the dominance of eastward current reduces, and the current tends to be dominantly westward. Furthermore, Figures 12a-c show the first mode of EOF vertical structure and its temporal variability (72% of total variance) at $C_{EJ}$. In general, anomaly of the zonal current at $C_{EJ}$ is westward, which is supposed to be associated with the meeting of SEC driven by trade winds and the ITF at this region. The EEMD analysis of the EOF1 of zonal current at $C_{EJ}$ also shows intraseasonal-interannual variabilities (Figs. 12d-g), where it is found that interannual timescale dominates the zonal current variation at $C_{EJ}$ (0.017 power per year).

To obtain a better understanding of the zonal current characteristics at $A_{WJ}$, $B_{SM}$, and $C_{EJ}$, we have summarized maximum energy density of zonal currents at intraseasonal, semiannual, annual, and interannual timescales that exists at each point based on power spectrum calculation in Figs. 10-12 (Table 3). It is shown that the zonal currents at $A_{WJ}$ have peak energies, which

are consecutively dominated by semiannual, intraseasonal, and annual signals, while interannual signal is weaker than them at this point. Furthermore, although semiannual and intraseasonal signals are dominant at $B_{SM}$, there is pronounced interannual variation of the zonal current at this point. In contrast, the zonal current variability at $C_{EJ}$ is dominated by interannual signal. Furthermore, based on the power spectrum calculation shown in Fig.12 (Table 3), it is found that intraseasonal variability of the SEC (zonal current at $C_{EJ}$) is also prominent (~0.012 power per year) in addition to the interannual signal (~0.017 power per year). Meanwhile, based on sea level anomaly data in the period of October 1992 to the end of 1998 (about 6 years), Feng and Wijffels (2002) suggested that the strongest intraseasonal variability in the SETIO occurs in the SEC during the July-September season and baroclinic instability seems to be the leading cause. On the other hand, in this study, we found that the strongest intraseasonal variability occurs in the SJC (zonal current at $A_{WJ}$). This different result seems due to differences in the length of data used in this study (64 years) and that in Feng and Wijffels (2002) (6 years). In addition, in this study, we analyzed intraseasonal variability from the signal of the EOF first temporal mode of zonal currents (accounting for 58%, 64%, and 72% of total variance at $A_{WJ}$, $B_{SM}$, and $C_{EJ}$, respectively), whereas Feng and Wijffels (2002) analyzed the intraseasonal variation from standard deviation of the 6-year sea level anomaly data based on the 100-day high-pass filtered altimeter data during the four seasons (January-March, April-June, July-September, and October-December). Moreover, some of the difference may also be due to the fact that altimeter data do not resolve coastal processes well. However, further study is required to address this issue.

### 3.3.2 Interannual Variations

In this study, it is found that the most energetic zonal current variations of EOF1 at $A_{WJ}$, $B_{SM}$, and $C_{EJ}$ exist at ~30 m depth (Figs. 10a, 11a, and 12a). To investigate exclusively the ocean currents at interannual timescale, lagged correlation analyses have been applied between the zonal currents at a depth of about 30 m at points $A_{WJ}$, $B_{SM}$, and $C_{EJ}$ and each of the climatic indices (e.g., ONI and DMI), as shown in Table 4. The ONI and DMI indices from 1950 to 2013 used in this study are shown in Fig. 13.

The analysis of lagged correlation shows that the currents at $B_{SM}$ and $C_{EJ}$ show positive correlations with the ONI, namely $r(18)=0.24$ and $r(4)=0.27$, respectively, with the 95% significance level approximately ±0.07, indicating that an El Niño (La Niña) event is favourable for an eastward (westward) currents at these points (Figs. 11g and 12g) and also pointing out that ITF transport is lower (higher) during El Niño (La Niña) events (Fieux et al., 1996; Meyers, 1996; Gordon and Susanto, 1999; Ffield et al., 2000; Susanto et al., 2001; Susanto and Gordon, 2005; Susanto et al., 2012; Liu et al., 2015; Susanto and Song, 2015; and Zhang et al., 2016). ENSO seems to have a strongest influence on the zonal current variability at $C_{EJ}$ (Table 4), which is located close to the exits of the ITF. The ENSO signals penetrate into the SETIO mainly through the equatorial Pacific and coastal ocean Indonesian waveguides (Wijffels and Meyers, 2004; Zhang et al., 2016). Meanwhile, the present study shows that the correlation between the zonal current at $A_{WJ}$ and ONI is weak and below the significance level.

Furthermore, negative correlation is found between IOD and zonal currents at $A_{WJ}$ [DMI−$U$: $r(9)=-0.09$], $B_{SM}$ [DMI−$U$: $r(1)=-0.28$], and $C_{EJ}$ [DMI−$U$: $r(11)=-0.13$]. The correlation analysis indicates that IOD is most influential to force interannual

variation of the zonal currents at $B_{SM}$, with the IOD leading the zonal currents by 1 month. The influence of interannual

phenomenon at $B_{SM}$, such as IOD, is stronger and relatively instantaneous than that at points $C_{EJ}$ and $A_{WJ}$. This may be due to the location of $B_{SM}$, which is close to the center of eastern pole of the IOD (100° E, 5° S; Saji et al., 1999). In contrast to ONI, there is IOD signals at $A_{WJ}$ although the IOD signals at this point are weak compared to $B_{SM}$, and $C_{EJ}$ (Table 4). This indicates that some of the IOD signals are coastally trapped.

Table 5 lists extreme and neutral years and their concurrent events through 1950-2013. To further investigate interannual

variation of zonal current, we summarized presence of major climate modes (ENSO and/or IOD) and the corresponding current anomalies at the points of $B_{SM}$ and $C_{EJ}$ (Table 6) based on the lagged correlation analyses in Table 4 and the interannual variations of zonal current (Figs. 11g and 12g), and the ONI and DMI (Fig. 13), respectively. In the Table 4, the ONI-*U* and DMI-*U* correlations are independent of IOD and of ENSO, respectively. Meanwhile, the current anomalies, which are attributed to the presence of major climate modes (ENSO and/or IOD) shown in the Table 6, could be forced by ENSO or

IOD, or the combined effect of them. In this study, the amounts of respective contribution values of ENSO and IOD, or the combined effect of them on the current anomalies shown in the Table 6 are still unknown. Further studies are thus required to more quantitatively determine the contribution values of each of climate modes on zonal current variations in the study area as well as their possible teleconnection.

In addition to the lagged correlation analysis (Table 4), partial correlation analysis was also conducted since the IOD tend to

co-occur with ENSO. Table 7 shows the partial correlation coefficients between zonal currents at 30 m on interannual timescale and each ONI and DMI. As for the ONI, the currents revealed significant positive correlations at $C_{EJ}$ during all monsoon seasons. This positive correlation suggests that El Niño (La Niña) events caused an eastward (westward) anomaly of currents at this point. Meanwhile, the partial correlation between the currents and the DMI showed significant negative correlation at $B_{SM}$, in which it occurred only during the SE monsoon (JJA), as shown in Table 7. This negative correlation indicates that an

eastward (westward) anomaly of the currents was induced by negative (positive) IOD. The results of the partial correlation analysis confirm and complement the previous findings in Table 4 that ENSO mainly contributed to the zonal current variability at $C_{EJ}$ in DJF, MAM, JJA, and SON, whereas the IOD had a significant influence on the variability of current at $B_{SM}$ and only in JJA. In this present study, however, what are the causes of the influence of IOD on the current variability at $B_{SM}$ only in JJA are still unresolved. Further research is necessary to explain the dynamical links of this matter. Additionally,

the last mode (Figs. 10h, 11h, and 12h) represents long-term variation and trend, which may be associated with long-term internal variability within the Indian Ocean or remote forcing from the Pacific Ocean and they may discuss in detail in future paper.

### 3.3.3 Relationship of the Zonal Current Variations at $A_{WJ}$, $B_{SM}$, and $C_{EJ}$ to Both Remote and Local Wind Forcings

To confirm possible influences of wind forcings on dominant variations of zonal current at $A_{WJ}$, $B_{SM}$, and $C_{EJ}$, we have

calculated the correlation between them. In this study, it is found that the zonal currents at $A_{WJ}$ (close to the shore) have peak energy at semiannual period (0.140 power per year; Table 3). The semiannual variations of the zonal current at $A_{WJ}$ show the

presence of an eastward anomaly of the zonal current during MAM and SON, which may be associated with Kelvin waves forced by winds over the equatorial Indian Ocean (Wyrtki, 1973; Quadfasel and Cresswell, 1992; Sprintall et al., 1999, 2000, 2010). Furthermore, we have calculated the correlation between zonal currents in the upper layer (30 m) at $A_{WJ}$ and zonal
winds for the semiannual signals extracted using the EEMD method (Fig. 15). The 30 m upper layer flows at $A_{WJ}$ show a strong positive correlation with the zonal winds over the equatorial Indian Ocean, with the winds leading the current by approximately one month. The positive correlation indicates that the flows are to the east when the winds blow from the west to the east, and vice-versa for the easterly wind. The one-month lag between the flows at $A_{WJ}$ and the zonal winds in the equatorial Indian Ocean is in agreement with the expected arrival time of Kelvin waves at this point, suggesting that it is of
about 18 – 35 days with phase speeds ranging from 1.5 to 2.9 m s$^{-1}$ (e.g., Sprintall et al., 2000; Syamsudin et al., 2004; Iskandar et al., 2005). Interestingly, there is also a weaker positive correlation of the 30 m upper layer flows at $A_{WJ}$ at lag of about one month with zonal trade winds in the western equatorial Pacific Ocean (WEPO) at semiannual timescale, indicating that a strengthening (weakening) of easterly trade winds over the WEPO is favourable for anomalous westward (eastward) currents at $A_{WJ}$. The strengthening of easterly trade winds over the WEPO will increase sea level in the northern waters of West Papua
and New Guinea, enhancing eastward pressure gradient across the Indonesian seas and forcing strengthened ITF transport. Since the currents at $A_{WJ}$ are strongly correlated to the ITF (Table 1), it is suggested that this possible dynamic could result in anomalous westward currents at $A_{WJ}$, and vice-versa for the weakening winds over the WEPO.

Semiannual (0.135 power per year) signal of current variations is also dominant at $B_{SM}$ but it is weaker than that at $A_{WJ}$. In addition, there is pronounced interannual (0.012 power per year) variation of the zonal current at $B_{SM}$ (Table 3 and Fig. 11g),
in which IOD is most influential to force interannual variation of currents at this point (at 30 m), as shown in Table 4. Like at $A_{WJ}$, we also look for the relationships of the upper layer flow (30 m) at $B_{SM}$ with the zonal winds but for the interannual signal obtained using the EEMD method (Fig. 16). At interannual timescale, the 30 m upper layer flows at $B_{SM}$ show a strong positive correlation with the zonal winds over the eastern tropical Indian Ocean, in which the response of the flows to the zonal winds are relatively instantaneously at a lag of about one month (Fig. 16). Location of the zonal winds affecting interannual variations
of the upper layer flows at $B_{SM}$ is in accord with the eastern pole region of IOD (90°E-110°E, 10°S-0°S; Saji et al. 1999). Furthermore, as already explained, the zonal current variability at $C_{EJ}$ (close to the exits of the ITF) is dominated by interannual (0.017 power per year) signal in which the influence of ENSO is strongest at this point at depth of 30 m (Table 4). To enhance our understanding of possible relationship of zonal currents at $C_{EJ}$ to wind forcings at interannual timescale, we have also calculated the correlation between the upper layer flow (30 m) at $C_{EJ}$ and the zonal winds, particularly in the Pacific Ocean.
Like at $B_{SM}$, the interannual signals of both flows and winds are extracted using the EEMD method. At interannual timescale, the flows at $C_{EJ}$ at 30 m show a significant positive correlation with the local winds and the remote winds over the equatorial Pacific Ocean, in which the response of the flows to the zonal winds are about 4 to 6 months. In addition, we also found that the 4-month lag signal is stronger than the signals with the 5 to 6 months of lag. Figure 17 shows a correlation map between the Pacific winds and the currents at $C_{EJ}$ in the case of a 4-month lag.

Previous study conducted by Wijffels and Meyers (2004) shows that the variability in the ITF region associated with Kelvin and Rossby waves originating in the Indian and Pacific Oceans, respectively. They have revealed the pathways for equatorial Pacific wind energy traveling down the Papuan/Australian shelf break and radiating westward-propagating Rossby Waves into the Banda Sea and southeast Indian Ocean (their Fig. 20). Hence, there is a contribution of the westward-propagating Rossby waves to the ITF variability inside the Indonesian seas or at the ITF exit regions (Ombai and Lombok Straits, and Timor

Passage), which lead into the SETIO as well as the western coast of Sumatra and the southern coast of Java. Our simulation (Fig. 2) clearly shows that ITF flowing from the exit passages of Indonesian seas (Lombok, Ombai, and Timor passages) feeds into the SETIO region. Moreover, Wijffels and Meyers (2004) have computed the remotely driven Pacific Rossby wave speeds as a function of latitude. The phase speeds have been compared with the theoretical Rossby wave speeds based on atlas of Chelton et al. (1998). In this study, we have estimated the travel time of the westward-propagating Rossby waves excited by

the wind anomalies in the central and western Pacific to the SETIO, especially at point $C_{EJ}$, based on the pathways for the Pacific signals introduced by Wijffels and Meyers (2004). In general, it was found that the equatorial Pacific signals around 130° W took approximately 3.01 months to the $C_{EJ}$ based on the mean phase speed of about 0.2 cm s$^{-1}$ taken from Wijffels and Meyers (2004). This travel time estimation was within the range of the 4-month lags between the flows at $C_{EJ}$ and the Pacific winds derived from the lagged correlation analysis in Fig. 17.

**4 Conclusions**

Basic features of zonal currents and their temporal variability in the SETIO region adjacent to the Sumatra-Java southern coasts have been studied using global HYCOM output over the course of 1950 – 2013. There exist peculiar features of zonal currents in coastal (the SJC) region, offshore (the ITF/SEC) region, and transition zone between coastal and offshore regions of the SETIO. In general, surface zonal currents on Transect A (the SJC region), especially along the southern coasts of Sumatra-

Java (98° E-114° E), show seasonal characteristics, which are eastward (westward) during DJF (JJA). Moreover, the eastward-flowing currents are enhanced during MAM and SON associated with the propagation of coastal Kelvin waves. On the other hand, westward currents are dominant along Transect C (the ITF/SEC region). Meanwhile, although westward currents are quite dominant along Transect B (the transition zone between the SJC region and ITF/SEC region), eastward currents are also present, especially on longitudes 95° E to 107° E.

In the period of 1950 through 2013, the mean (climatological) current velocity of SJC on Transect A is dominantly eastward. We found that both remote and local wind forcings as well as seasonal conditions are necessary to explain the current variability in the study area. During JJA, the strength of climatological eastward SJC reduced and the SJC in the upper 100 m along the southern coast of Java, at a certain period of time, flowed westward in response to the prevailing southeasterly local winds during those months. At the depth 100 m, there is a maximum westward current at $A_{WJ}$ during DJF with velocity of about 0.1

m s$^{-1}$, in which the current at the $A_{WJ}$ shows high correlation with the subsurface (200 m) maximum ITF in the southern Ombai Strait (remote forcing), whereas its correlation with the NW local wind is weak. Otherwise, it is found that the NW zonal wind

is more influential to force variation of zonal current at $A_{EJ}$ than the ITF. Therefore, it is suggested that the westward current simulated at $A_{WJ}$ at 100 m during DJF is ITF-related, whereas that at $A_{EJ}$ at 100 m is relatively NW zonal wind-related.

Moreover, it is found that the average (climatological) current at $B_{SM}$ is eastward, while at points $B_{WJ}$ and $B_{EJ}$ it is westward, suggesting that the mean eastward current at $B_{SM}$ is influenced by tropical current systems in the Indian Ocean, such as the ECC, SWMC, and Wyrtki Jet, whereas the mean westward currents at the points $B_{WJ}$ and $B_{EJ}$ are more dominated by the ITF. In contrast, current characteristics on Transect C (offshore region) generally show similarities at all points ($C_{SM}$, $C_{WJ}$, and $C_{EJ}$), where the current along this transect flows westward throughout the year, confirming that Transect C is the SEC/ITF region. Seasonal variation of the westward current on the Transect C agrees well with that of ITF in Lombok Strait, Timor Passage, and through the three exit passages (the total ITF through the Lombok and Ombai Straits, and Timor Passage), in which during JJA, the flow is stronger than during DJF.

The EOF1 mode of zonal current across the three meridional sections (EJ, WJ, and SM) clearly shows unique features of zonal currents between nearshore and offshore regions in the sections. In Sections EJ and WJ, the vertical structure of EOF1 is characterized by one-layer flow. In the nearshore area of Section EJ, the vertical structure of EOF1 displays a gradual decrease in speed from the surface to 800 m depth, whereas in the transition and the offshore areas the flow velocities decline more rapidly with depth, reducing to nearly zero at depths of about 500 m and 300 m, respectively. Meanwhile, in Section WJ, the one-layer flow of the vertical structure of EOF1 shows a unidirectional vertical structure, which gradually decreases from the surface to a depth of ~450 m in all areas. On the contrary, in the nearshore and transition regions of Section SM, it is marked by two-layer flow, in which the velocity reversal between the two types of flow taking place at a depth of approximately 100 m and 200 m, respectively. Meanwhile, in the offshore area of Section SM, the vertical structure of EOF1 exhibits a unidirectional flow from the surface to a depth of about 500 m.

In this study, the predominant variation content of the zonal current anomalies in the region is quantitatively identified, varying from intraseasonal to interannual timescales. The analysis indicates that the zonal currents at $A_{WJ}$ (close to the shore) have peak energies, which are successively dominated by semiannual, intraseasonal, and annual periods, in which interannual period is weaker than them at this point. Moreover, although semiannual and intraseasonal variations are dominant at $B_{SM}$ (close to the center of eastern pole of the IOD), there is pronounced interannual variation of the zonal current at this point. In contrast, the zonal current variability at $C_{EJ}$ (close to the major exit passages of the ITF) is dominated by interannual signal. Nevertheless, in addition to the interannual signal, the power spectrum analysis shows that intraseasonal variability of the zonal current (SEC) at $C_{EJ}$ is also prominent. The lagged correlation analysis shows that ENSO seems to have a strongest influence on the zonal current variability at $C_{EJ}$, with the zonal current lagging the ENSO by 4 months. Meanwhile, the IOD is most dominant in controlling interannual fluctuation of the zonal current at $B_{SM}$, with the IOD leading the zonal currents by 1 month. Furthermore, based on the partial correlation analysis, it has been revealed that ENSO contributes to the zonal current variation at $C_{EJ}$ in all monsoon seasons (DJF, MAM, JJA, and SON), while the IOD plays a significant role in controlling the variation of current at $B_{SM}$ only in JJA. In this study, the dynamical links that cause the influence of IOD on the current variability at $B_{SM}$ only in JJA are still not known. Therefore, further study is essential to elucidate the physical mechanisms responsible for

this topic. In this study, it might be able to resolve variations on time scales greater than interannual cycle based on the HYCOM output over a relatively long period of time. Here, the proportion calculation of contribution of each EEMD mode to the EOF1 showed that the order of each mode's contribution from largest to smallest at $A_{WJ}$ and $B_{SM}$ is intraseasonal, semiannual, annual, interannual, and long-term signals. Interestingly, the contribution of long-term signal at $C_{EJ}$ is larger than the interannual and

annual signals. However, the detailed analysis of long-term signal is not the scope of this research and might be considered as future study. Moreover, future works, which include detailing the forcing mechanisms as well as investigating decadal variability and determining the cause of the long-term signals, are necessary to be performed to gain a better understanding of these interesting topics.

**Data Availability**

The data used in this study are deposited at https://www.oceanography.fitb.itb.ac.id/member/nsn/

**Author Contribution**

Primary author NSN formulated research goals and aims, developed methodology, conducted investigation process, designed model, and prepared the published work. SLS maintained research data, prepared data presentation, and drafted the initial manuscript. RDS supervised the research project and EEMD methodology. FH designed model simulation and validated the

model results.

**Competing Interest**

The authors declare that they have no conflict of interest.

**Acknowledgements**

The authors would like to gratefully acknowledge data support from the following institutions. The moored RAMA current is

provided by the NOAA (https://www.pmel.noaa.gov/tao/data_deliv/deliv-nojava-rama.html), while the INSTANT current is available at the INSTANT Web site (http://www.marine.csiro.au/~cow074/instantdata.htm). The wind fields are obtained from the NOAA (https://www.esrl.noaa.gov/psd/data/gridded/data.ncep.reanalysis.derived.surface.html). The ONI is provided by the NOAA/CPC at http://www.cpc.ncep.noaa.gov/data/indices/. The DMI is from the JAMSTEC at http://www.jamstec.go.jp/frcgc/research/d1/iod/iod/dipole_mode_index.html. Meanwhile, the HYCOM simulation results are

available at Research Group of Oceanography-ITB, Faculty of Earth Sciences and Technology, Bandung Institute of Technology (https://www.oceanography.fitb.itb.ac.id/member/nsn/). We would like to thank the support given by the DIKTI under Basic Research Grant 2019-2021 and Program of World Class Professor (WCP) 2018, for funding this works

and making the writing of this paper possible. R. Dwi Susanto is supported WCP-2018 and National Aeronautics and Space Administration (NASA) grant #80NSSC18K0777 and Jet Propulsion Laboratory-NASA subcontract #1554354 to the University of Maryland. We appreciate the valuable suggestions, comments, and corrections from anonymous reviewers.

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

**Table 1**. Correlation coefficients between zonal currents at 100 m depth at both $A_{WJ}$ and $A_{EJ}$ and each the local NW zonal wind and subsurface (200 m) maximum ITF in the southern Ombai Strait during DJF in the period of 2004 through 2006.

| Points | Correlation Coefficients $(r)$[a] | |
|:---:|:---:|:---:|
| | $U$-SMITF | $U$-NWZW |
| $A_{WJ}$ | 0.76 | -0.32[b] |
| $A_{EJ}$ | -0.13[b] | 0.30[b] |

[a] The 95% significance level is approximately $\pm 0.33$. $U$: zonal currents at 100 m depth; SMITF: subsurface (200 m) maximum ITF in the southern Ombai Strait; NWZW: northwesterly zonal wind.

[b] Correlation below the significance level.

**Table 2. Dominant variances at the nine observation points.**

| Mode | Variance (%) | | | | | | | | |
|:---:|:---:|:---:|:---:|:---:|:---:|:---:|:---:|:---:|:---:|
| | Section EJ | | | Section WJ | | | Section SM | | |
| | $A_{EJ}$ | $B_{EJ}$ | $C_{EJ}$ | $A_{WJ}$ | $B_{WJ}$ | $C_{WJ}$ | $A_{SM}$ | $B_{SM}$ | $C_{SM}$ |
| 1 | 60 | 76 | **72** | **58** | 84 | 87 | 37 | **64** | 88 |
| 2 | 29 | 18 | 20 | 33 | 12 | 10 | 25 | 27 | 9 |
| 3 | 6 | 4 | 3 | 5 | | | 13 | 6 | |
| 4 | | | 2 | | | | 10 | | |
| 5 | | | | | | | 6 | | |
| 6 | | | | | | | 4 | | |
| **Total** | **95** | **98** | **97** | **96** | **96** | **97** | **95** | **97** | **97** |

**Table 3. Maximum energy density (peak energies) at intraseasonal, semiannual, annual, and interannual timescales at points $A_{WJ}$, $B_{SM}$, and $C_{EJ}$.**

| Points | Maximum Energy Density (power per year) and Periods (months) | | | |
|:---:|:---:|:---:|:---:|:---:|
| | IS | SA | AN | IA |
| $A_{WJ}$ | 0.070 (3.0) | 0.140 (6.0) | 0.038 (12.0) | 0.003 (36.0) |
| $B_{SM}$ | 0.015 (3.0) | 0.135 (6.0) | 0.007 (12.0) | 0.012 (36.0) |
| $C_{EJ}$ | 0.012 (2.0) | 0.008 (6.6) | 0.012 (12.0) | 0.017 (44.4) |

IS: Intraseasonal; SA: Semiannual; AN: Annual; IA: Interannual. Values shown in brackets are periods.

**Table 4. Lag correlation between the zonal currents at 30 m and each ONI and DMI.**

| Points | Correlation Coefficients ($r$)[a] and Time Lag (TL) | | | |
|---|---|---|---|---|
| | ONI – $U$ | | DMI - $U$ | |
| | $r$ | TL (months) | $r$ | TL (months) |
| $A_{WJ}$ | 0.02[b] | 2 | -0.09 | 9 |
| $B_{SM}$ | 0.24 | 18 | -0.28 | 1 |
| $C_{EJ}$ | 0.27 | 4 | -0.13 | 11 |

[a] The 95% significance level is approximately ± 0.07. $U$: zonal currents at 30 m. Positive correlation coefficients between the currents and the ONI indicate existence of an eastward (westward) anomaly of the currents during El Niño (La Niña). Meanwhile, negative correlation coefficients between the currents and the DMI indicate existence of an eastward (westward) anomaly of the currents during negative (positive) IOD. A positive (negative) lag indicates that the variability in a former variable (e.g., ONI or DMI) leads (lags) that in the latter variable (the zonal current).

[b] Correlation below the significance level.

**Table 5. ENSO, IOD, and neutral events during the 1950 − 2013 periods.**

| | El Niño | | | NR-ENSO | | | La Niña | | |
|---|---|---|---|---|---|---|---|---|---|
| P-IOD | 1951 | 1953 | 1963 | 1962 | 1967 | 1990 | 1970 | 1976 | 1985 |
| | 1965 | 1966 | 1969 | 2003 | 2013 | | 1999 | 2000 | 2006 |
| | 1972 | 1977 | 1982 | | | | 2007 | 2008 | 2010 |
| | 1983 | 1986 | 1987 | | | | 2011 | | |
| | 1991 | 1993 | 1994 | | | | | | |
| | 1997 | 2002 | 2004 | | | | | | |
| | 2009 | 2012 | | | | | | | |
| NR-IOD | | | | 1952 | 1957 | 1961 | 1950 | 1971 | 1973 |
| | | | | 1979 | 2001 | 2005 | 1974 | 1988 | 1989 |
| | | | | | | | 1995 | | |
| N-IOD | 1968 | 1992 | | 1956 | 1958 | 1959 | 1954 | 1955 | 1964 |
| | | | | 1960 | 1978 | 1980 | 1975 | 1984 | 1998 |
| | | | | 1981 | 1996 | | | | |

NR-ENSO: neutral ENSO (-0.5 °C < ONI < +0.5 °C); El Niño (ONI > +0.5 °C); La Niña (ONI < -0.5 °C); P-IOD: Positive IOD (DMI > +0.36 °C); NR-IOD: neutral IOD (-0.36 °C < DMI < +0.36 °C); N-IOD: negative IOD (DMI < -0.36 °C). The classification of ENSO events is determined by ONI (http://www.ESRL.noaa.gov/). Meanwhile, DMI is used for the classification of IOD events, with criterion according to Yuan et al. (2008).

**Table 6. Summary of major climate modes (ENSO and/or IOD) and the corresponding current anomalies through 1950 − 2013.**

| Points | Events | Zonal Current ($U$) | |
| --- | --- | --- | --- |
| | | Current Speed (m s$^{-1}$) | Observation Time |
| | NR-ENSO (Jan. 2004) and NR-IOD (Jun. 2005) | -0.21 | Jul. 2005 |
| | NR-ENSO (Dec 1980) and P-IOD (May 1982) | -0.19 | Jun. 1982 |
| | NR-ENSO (Aug. 1962) and N-IOD (Jan. 1964) | -0.28 | Feb. 1964 |
| | El Niño (Feb. 1998) and P-IOD (Jul. 1999) | -0.35 | Aug. 1999 |
| B$_{SM}$ | El Niño (Oct. 2009) and P-IOD (Apr. 2011) | -0.18 | May 2011 |
| | La Niña (Dec. 1995) and P-IOD (Jul. 1997) | -0.50 | Aug. 1997 |
| | La Niña (Aug. 2007) and P-IOD (Feb. 2009) | -0.16 | Mar. 2009 |
| | La Niña (Feb. 1995) and N-IOD (Jul. 1956) | -0.24 | Aug. 1956 |
| | La Niña (Oct. 1955) and NR-IOD (Mar. 1957) | -0.13 | Apr. 1957 |
| | NR-ENSO (Oct. 2001) and NR-IOD (Feb. 2001) | -0.18 | Jan. 2002 |
| | NR-ENSO (May. 1978) and P-IOD (Oct. 1977) | -0.46 | Sep. 1978 |
| | NR-ENSO (Mar. 1960) and N-IOD (Aug. 1959) | 0.41 | Jul. 1960 |
| | El Niño (Aug. 1953) and P-IOD (Jan. 1953) | 0.96 | Dec. 1953 |
| | El Niño (Nov. 1991) and P-IOD (Apr. 1991) | 0.45 | Mar. 1992 |
| | El Niño (Nov. 2009) and P-IOD (Apr. 2009) | 0.69 | Jan. 2010 |
| C$_{EJ}$ | El Niño (Jul. 1997) and N-IOD (Dec. 1996) | 0.78 | Nov. 1997 |
| | La Niña (May. 1988) and P-IOD (Oct. 1987) | -0.46 | Sep. 1988 |
| | La Niña (Sep. 1998) and P-IOD (Feb. 1998) | -0.59 | Jan. 1999 |
| | La Niña (Nov. 2011) and P-IOD (Apr. 2011) | -0.61 | Mar. 2012 |
| | La Niña (Aug. 1954) and NR-IOD (Jan. 1954) | -0.70 | Dec. 1954 |
| | La Niña (Sep. 1988) and NR-IOD (Feb. 1988) | -0.43 | Jan. 1989 |

NR-ENSO: neutral ENSO; P-IOD: Positive IOD; NR-IOD: neutral IOD; N-IOD: negative IOD. The classification criterion for ENSO and IOD events can be seen in Table 5.


**Table 7. Partial correlation coefficients between zonal currents at 30 m on interannual timescale and each ONI and DMI. Only values above 95% confidence level are shown.**

| Points | ONI – $U$ (no DMI) | | | | DMI – $U$ (no ONI) | | | |
|---|---|---|---|---|---|---|---|---|
| | DJF | MAM | JJA | SON | DJF | MAM | JJA | SON |
| $A_{WJ}$ | - | - | - | - | - | - | - | - |
| $B_{SM}$ | - | - | - | - | - | - | -0.76 | - |
| $C_{EJ}$ | 0.46 | 0.28 | 0.47 | 0.43 | - | - | - | - |
| The 95% significance level | ±0.19 | ±0.25 | ±0.26 | ±0.23 | ±0.63 | ±0.49 | ±0.35 | ±0.41 |







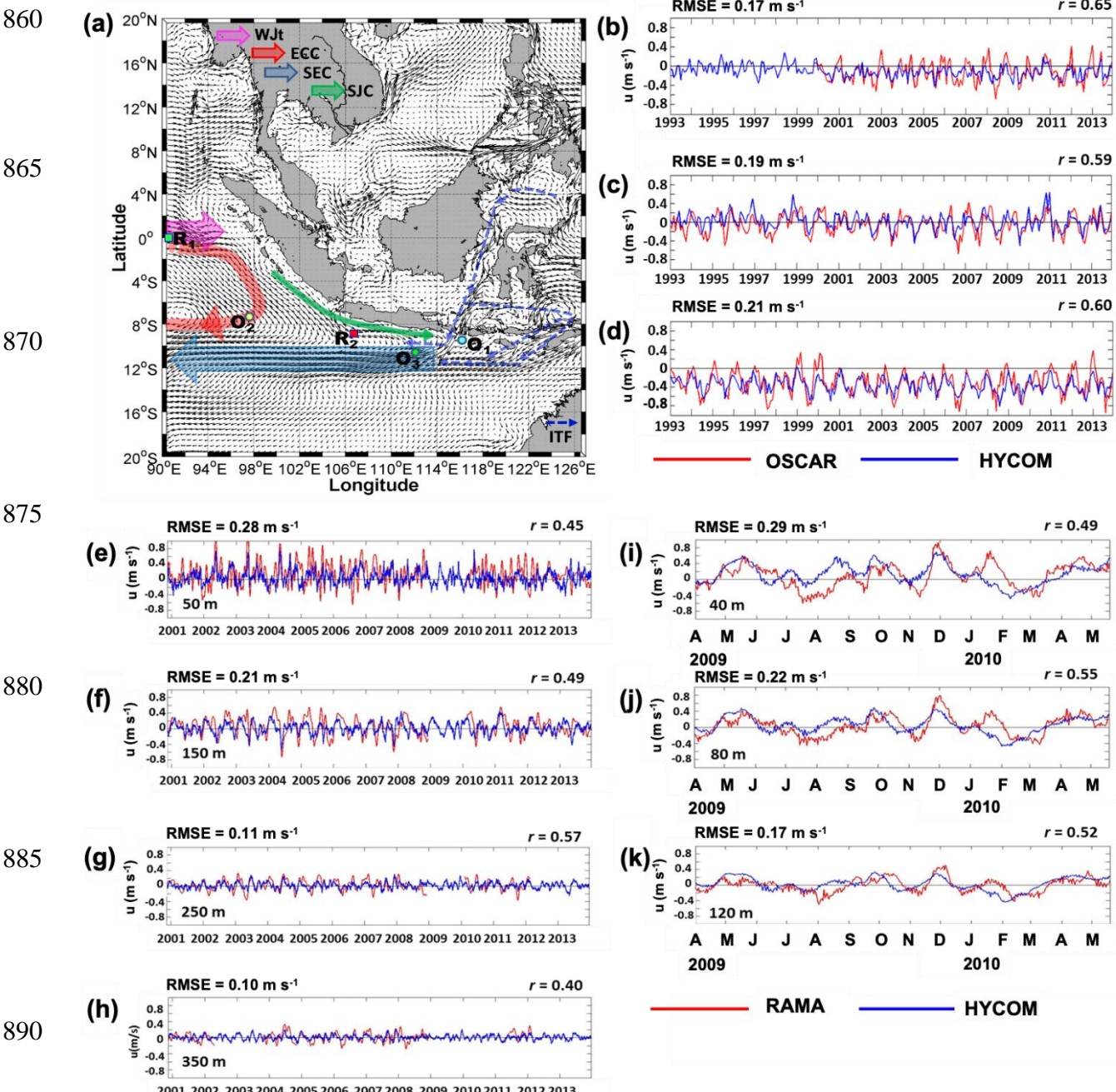

**Figure 1. Validation of HYCOM zonal currents with OSCAR and RAMA datasets: (a) Locations of validation points: Points O$_1$ (8ºS, 116ºE), O$_2$ (7ºS, 98ºE), and O$_3$ (11.5ºS, 113ºE) for the OSCAR data, while R$_1$ (0ºS, 90ºE) and R$_2$ (8.5ºS, 106.75ºE) for the RAMA data. (b)-(d) Time series of the zonal currents observed by the HYCOM (blue lines) and the OSCAR (red lines) at a depth of 0.5 m at point O$_1$, O$_2$, and O$_3$, respectively. Meanwhile (e)-(h) are the time series of zonal currents observed by the HYCOM (blue lines) and the moored RAMA (red lines) at point R$_1$ at depths of 50, 150, 250, and 350 m, sequentially.**

Meanwhile, (i)-(k) are the same as (e)-(h), except for point $R_2$ and depths of 40, 80, and 120 m, respectively. In the Figs. 1e-h (point $R_1$), a monthly low-pass filter has been applied before plotting. RMSE: root mean square errors; $r$: correlation coefficients.

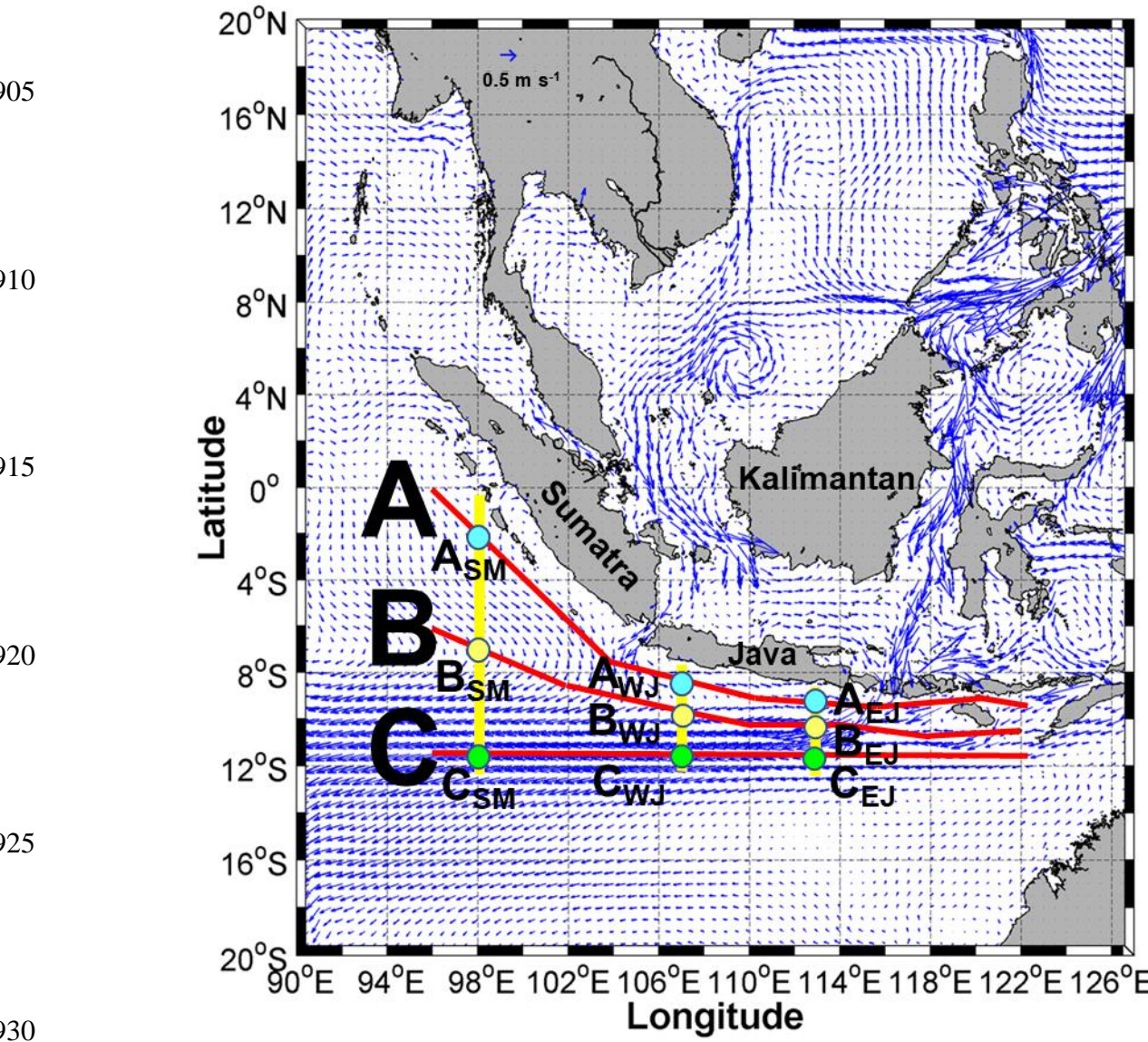

**Figure 2. The area of study interest in the SETIO region adjacent to the Sumatra-Java southern coasts. The blue arrows show climatological (yearly mean) surface (1 m) current field over 64 years from 1950 to 2013. Yellow lines are the meridional sections along the three longitudes (98°E, 107°E, and 113°E), while red lines are the three selected transects: A, B, and C. Green, yellow, and cyan circles are the locations in which the zonal currents are analyzed, namely points $A_{SM}$, $A_{WJ}$, $A_{EJ}$**

(on the Transect A); points B$_{SM}$, B$_{WJ}$, and B$_{EJ}$ (on the Transect B); and points C$_{SM}$, C$_{WJ}$, and C$_{EJ}$ (on the Transect C). The

subscripts SM, WJ, and EJ denote regions which close to Sumatra, West Java, and East Java.

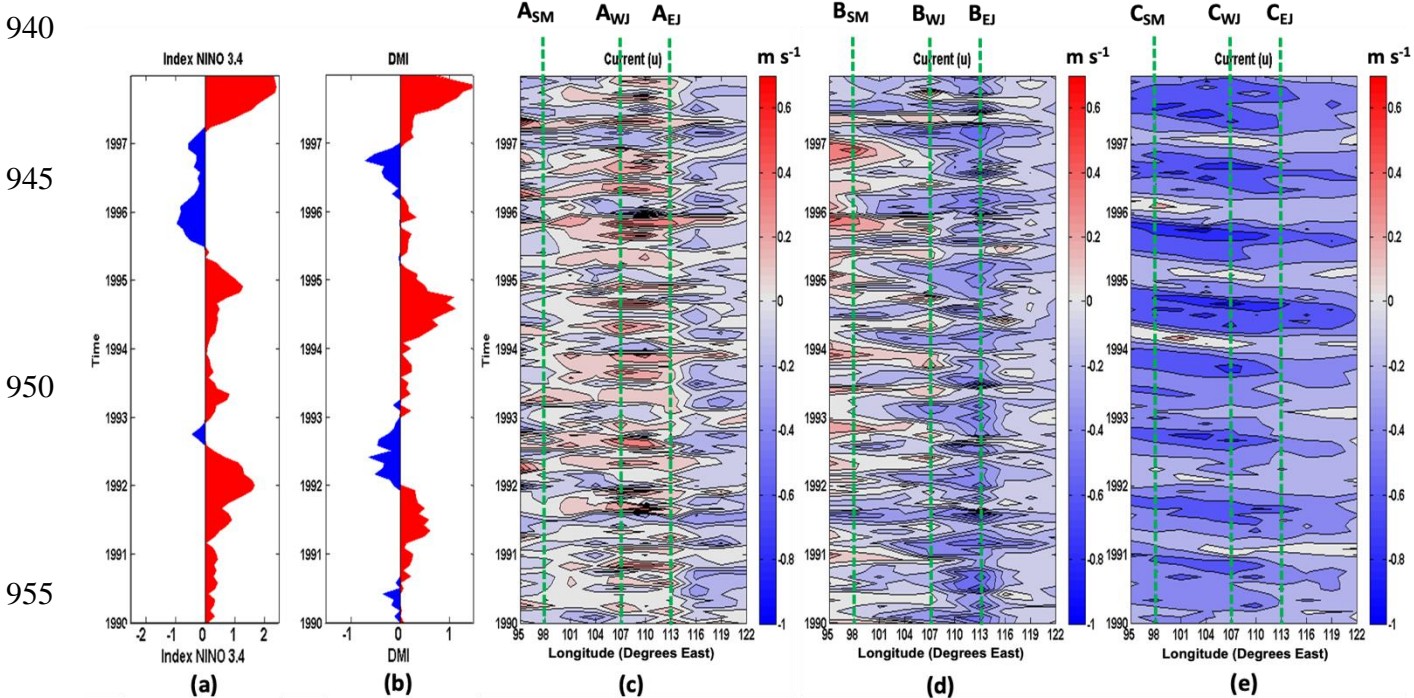

**Figure 3.** Time-longitude profiles of: (a) ONI, (b) DMI; and monthly averages of surface (1 m) zonal currents along (c) Transect A,
(d) Transect B, and (e) Transect C. Positive (negative) values of the zonal currents indicate eastward (westward).
Meanwhile, green dash lines denote longitudes of the nine selected points.

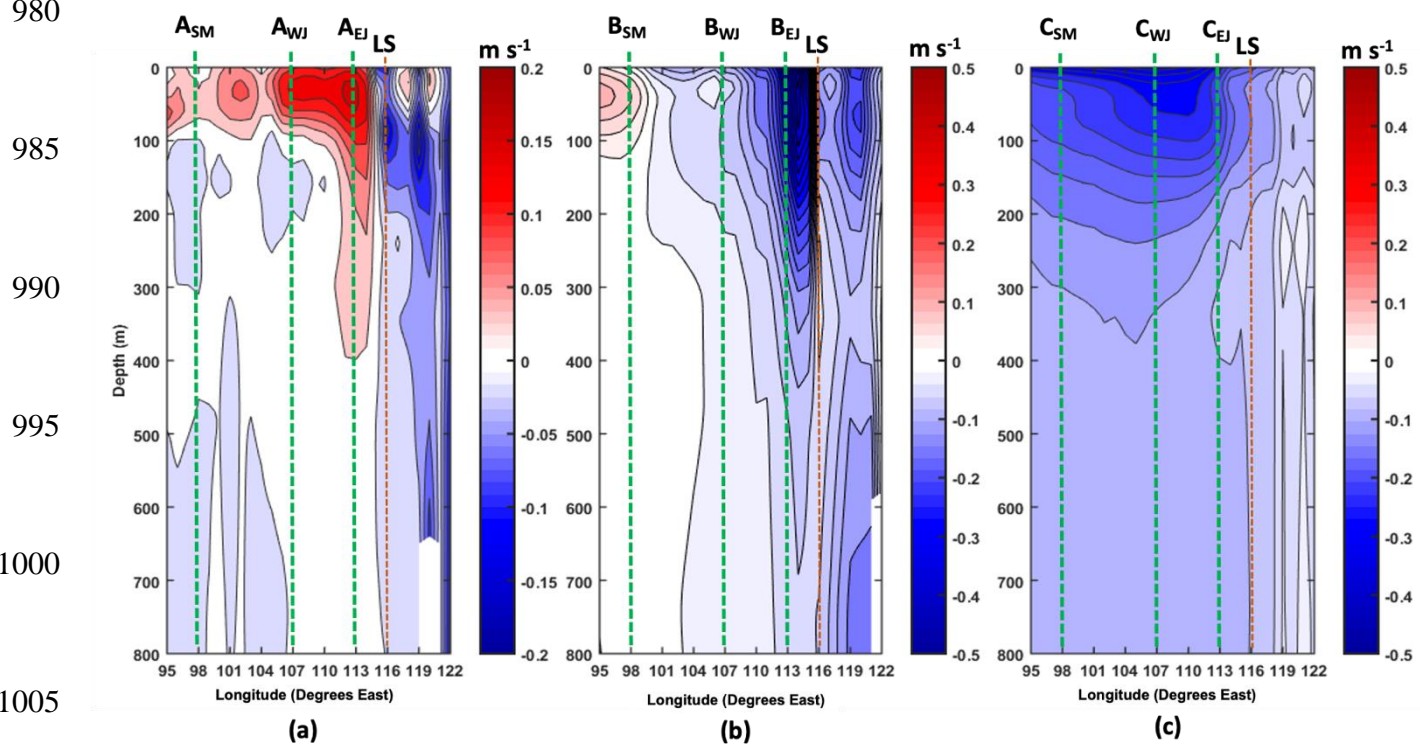

**Figure 4. Longitude-depth profiles of mean zonal currents along (a) Transect A, (b) Transect B, and (c) Transect C. Positive (negative) values of the zonal currents indicate eastward (westward). Green dash lines denote longitudes of the nine selected points, whereas dark orange dash lines denote longitude of Lombok Strait (LS).**

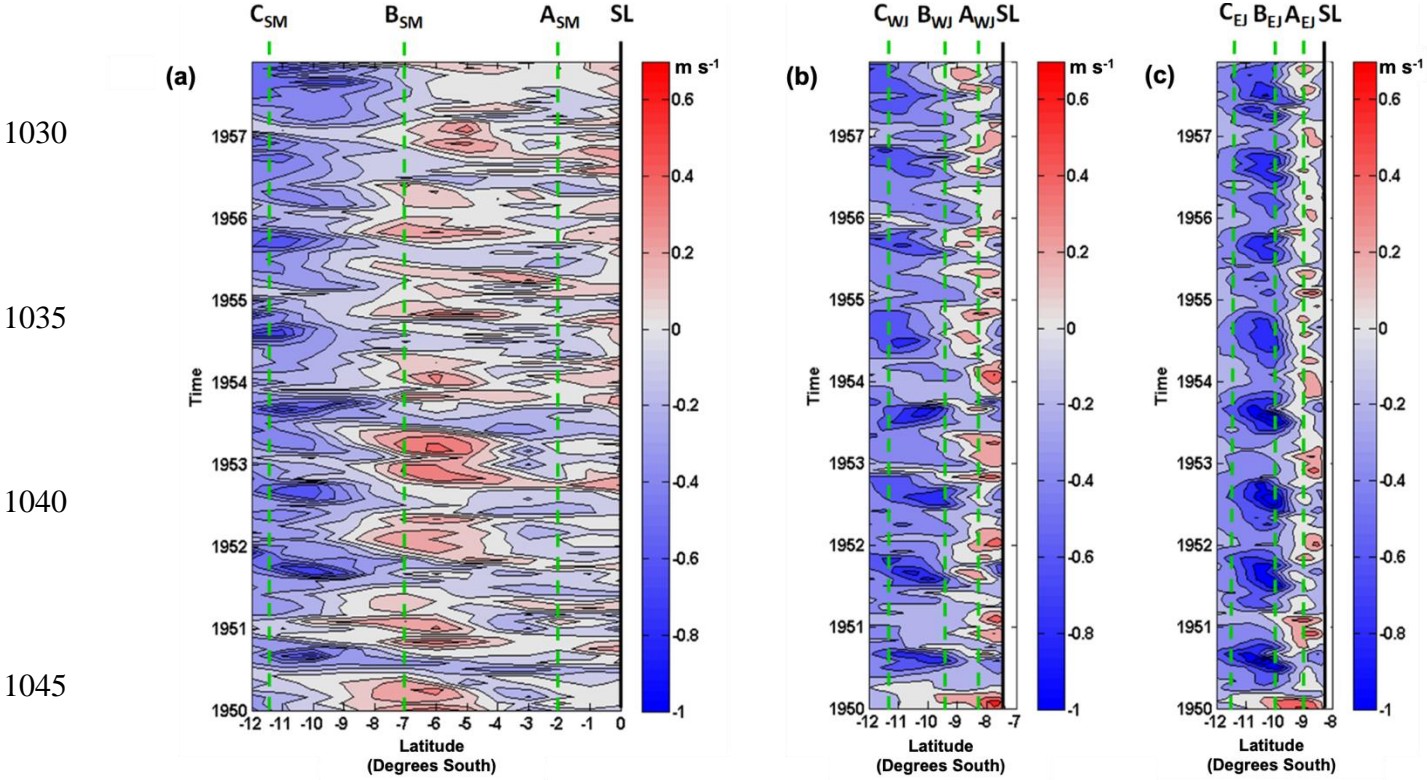

**Figure 5.** The zonal surface (1 m) currents along three meridional sections (yellow lines in Fig. 2): (a) SM (98ºE), (b), WJ (107ºE), and (c) EJ (113ºE). Positive (negative) values of the zonal currents indicate eastward (westward). Meanwhile, green dash lines denote latitudes of the nine selected points and SL is shoreline.

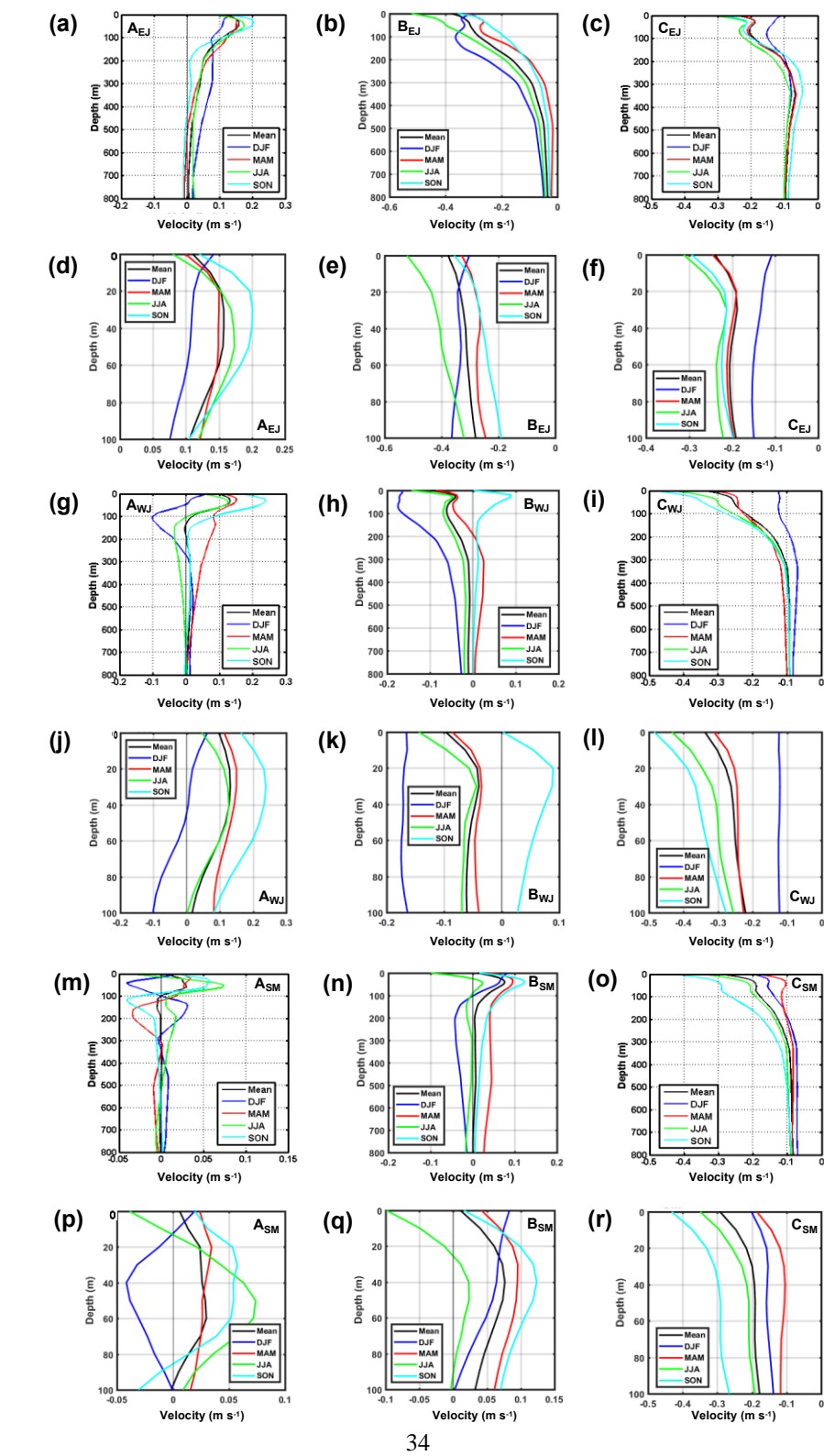

**Figure 6. Mean and seasonal profiles of zonal current velocity derived from the HYCOM simulation results for the period of 1950 through 2013, at points: (a) A_EJ, (b) B_EJ, (c) C_EJ, (g) A_WJ, (h) B_WJ, (i) C_WJ, (m) A_SM, (n) B_SM, and (o) C_SM. Meanwhile, (d)-(f), (j)-(l), and (p)-(r) are the same as (a)-(c), (g)-(i), and (m)-(o), respectively, except for depths of 0-100 m.**

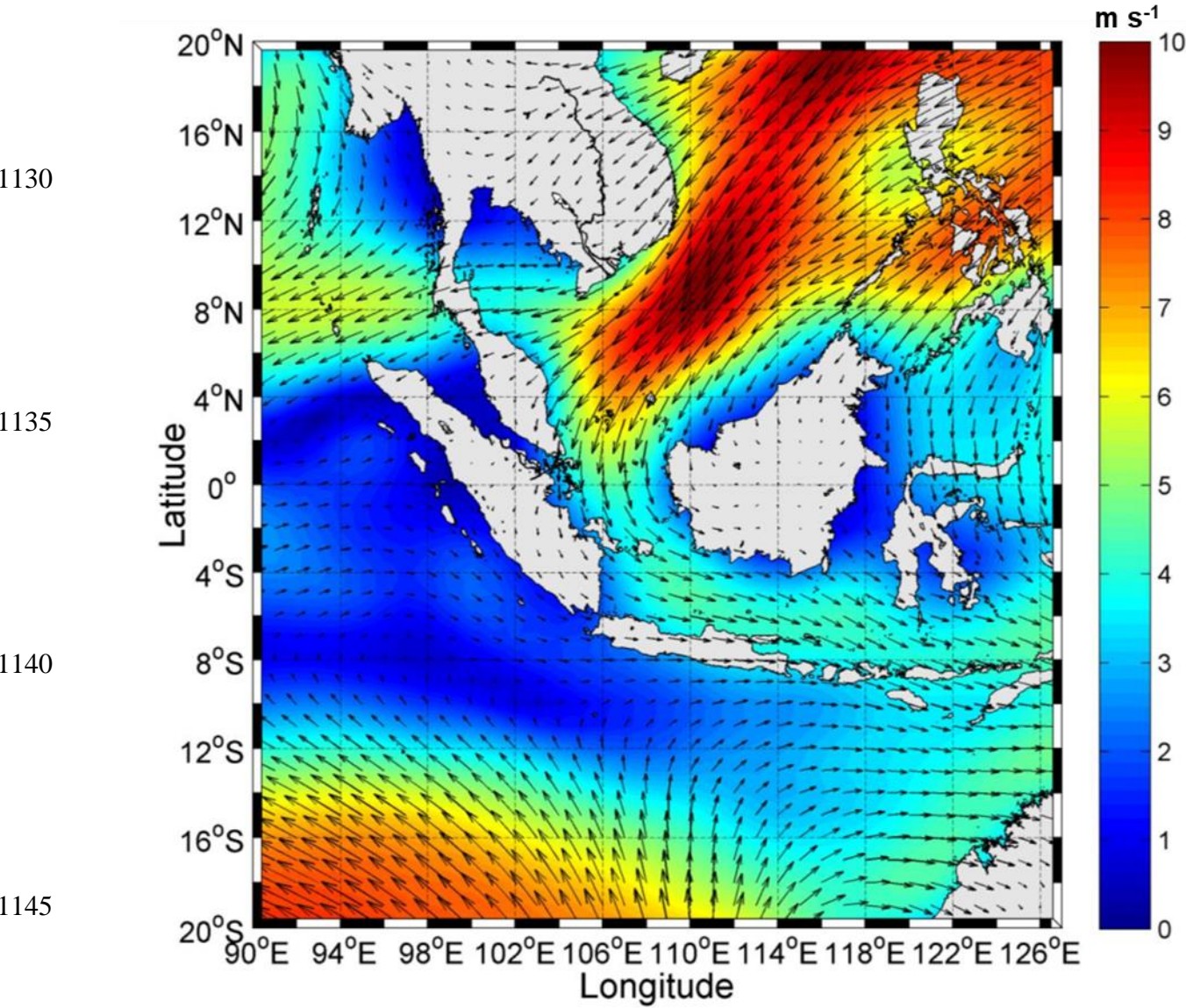

**Figure 7. Mean NW monsoon for the period of 1950 to 2013 (climatological wind field during the DJF).**

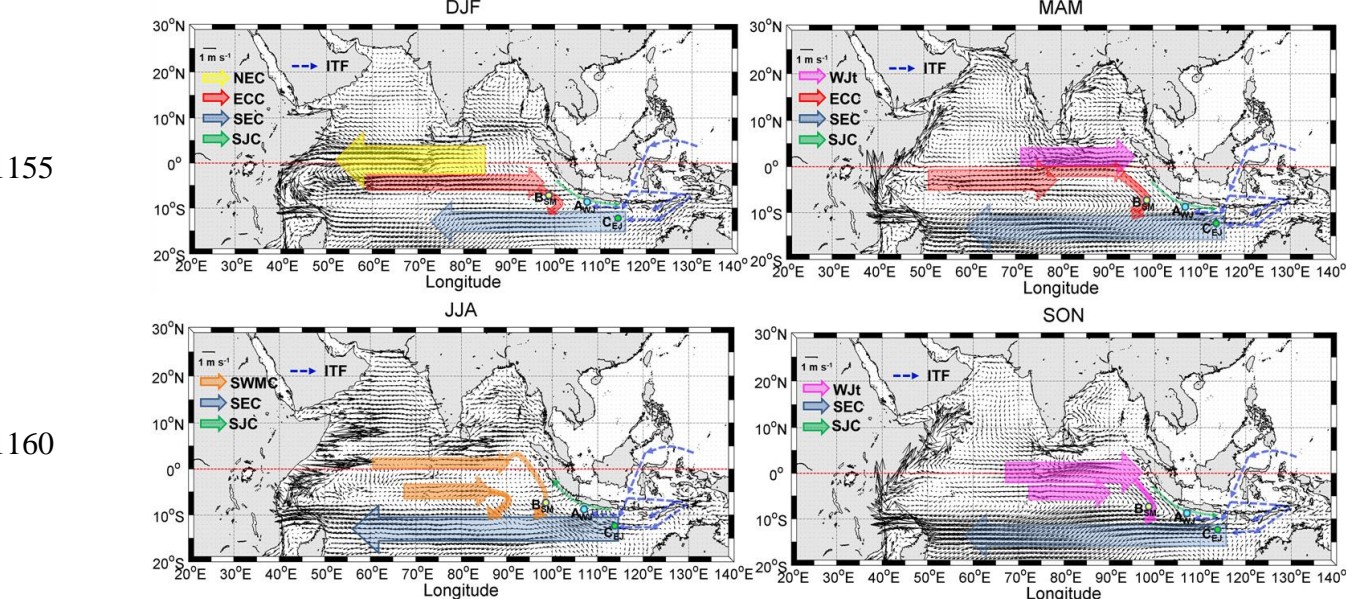

Figure 8. Seasonal averaged surface (1 m) currents over 64 years (1950-2013) and schematics of the tropical current systems in the Indian Ocean during (a) DJF, (b) MAM, (c) JJA, and (d) SON. Current branches indicated by colour arrows (not black) are the North Equatorial Current (NEC), Equatorial Counter Current (ECC), South Equatorial Current (SEC), South Java Current (SJC), Wyrtki Jet (WJt), South West Monsoon Current (SWMC), and Indonesian Throughflow (ITF). The dashed line represents thermocline current.

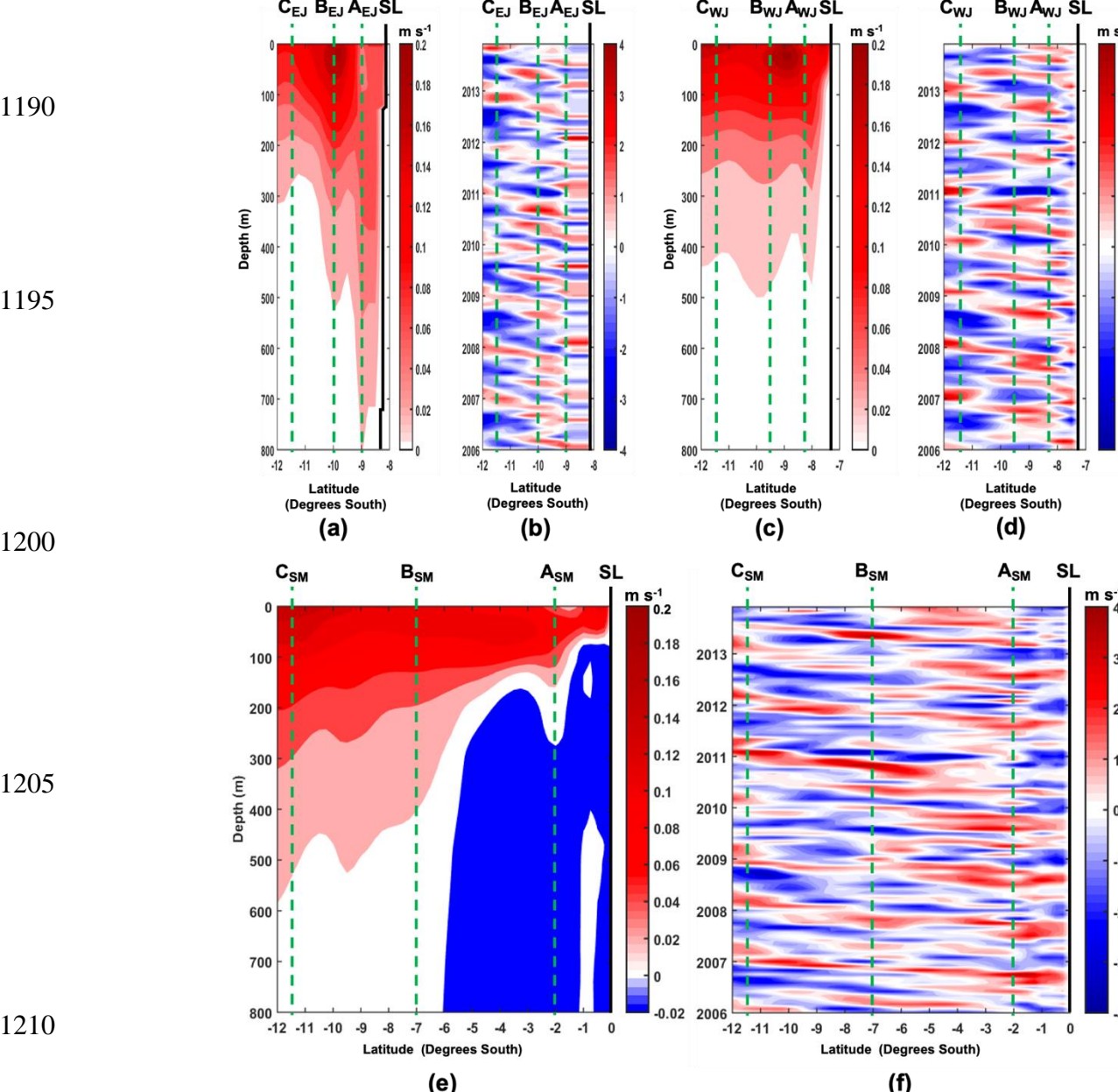

**Figure 9.** Vertical mode structures (a, c, and e) and their associated temporal variability of EOF1 (b, d, and f) of zonal currents relative to the mean flow along the three meridional sections: EJ (a and b), WJ (c and d), and SM (e and f). In this case, the temporal variability is shown for the last eight-year period of the EOF1. The direction of mode velocities relative to the mean flow is determined by multiplying the sign of the vertical mode structure and the sign of the temporal mode variability. Positive (negative) values of the velocity variability relative to the mean flow indicate eastward (westward). Meanwhile, green dash lines indicate latitudes of the nine analyzed points and SL is shoreline.

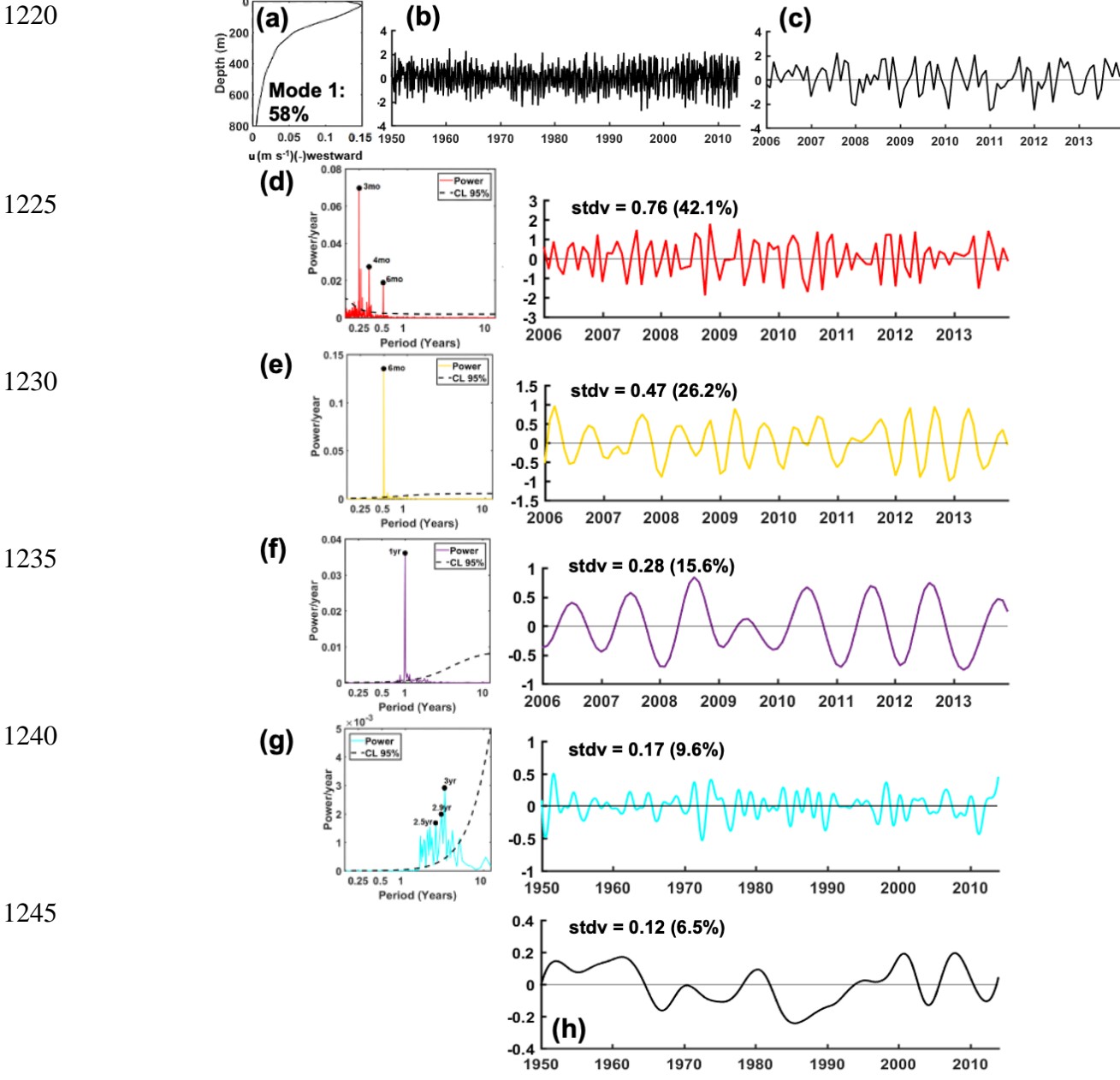

**Figure 10.** (a) Vertical mode structure and (b) its associated temporal variability of EOF1 (58% of total variance) at the point A_WJ.
(c) As (b), except for the last eight-year period of the EOF1. The EEMD is then applied to the EOF temporal structure to
decompose temporal variability: (d) intraseasonal, (e) semiannual (f) annual, and (g) interannual variabilities with their

corresponding red spectrum as a reference for 95% confidence limit (left panel), whereas (h) represents the long-term variation and trend. Meanwhile, stdv is standard deviation of each EEMD mode relative to the total variance of the EOF1.

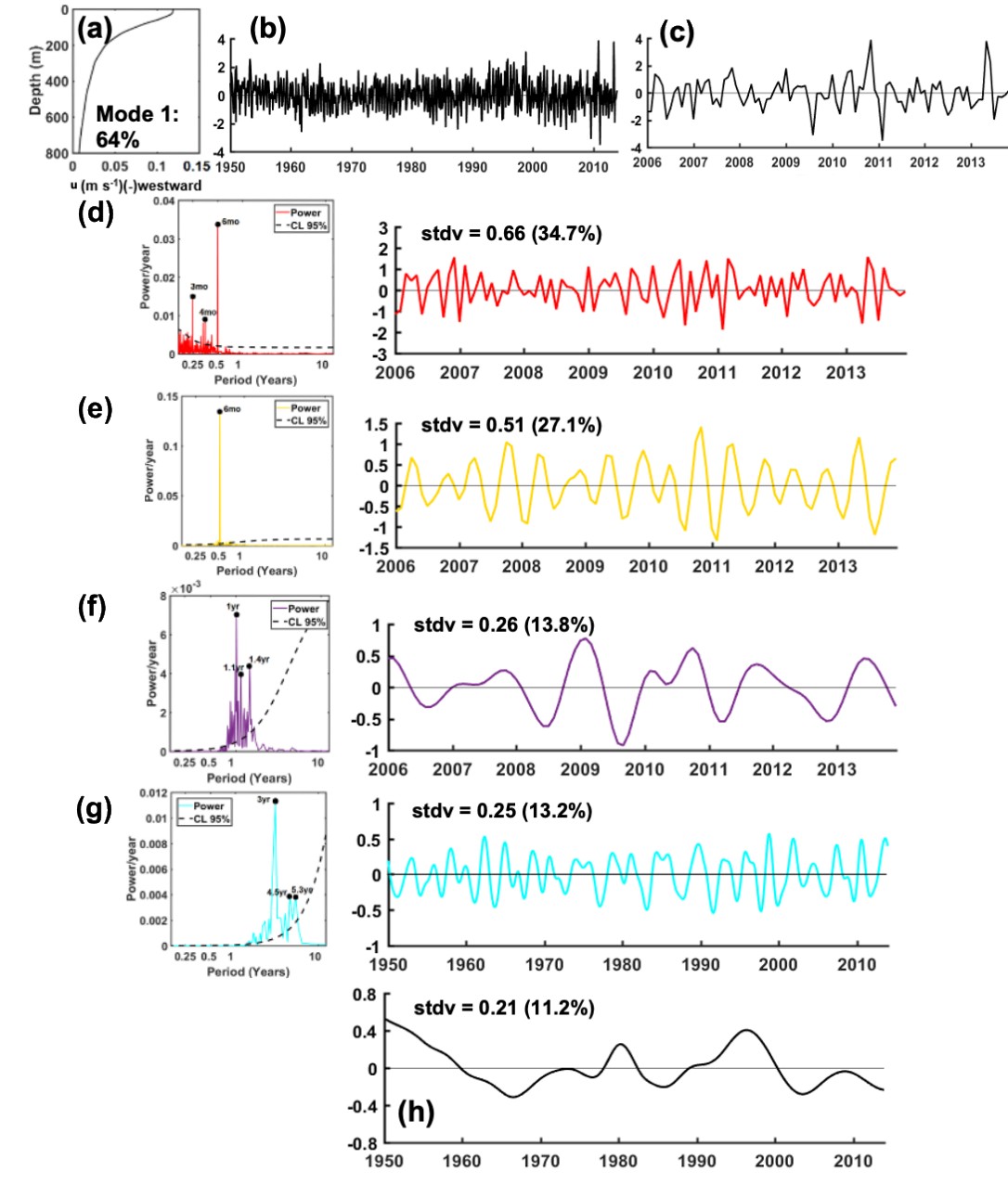

**Figure 11. Same as in Fig. 10, except for the point B$_{SM}$ with the temporal variability of EOF1 accounting for 64% of total variance.**

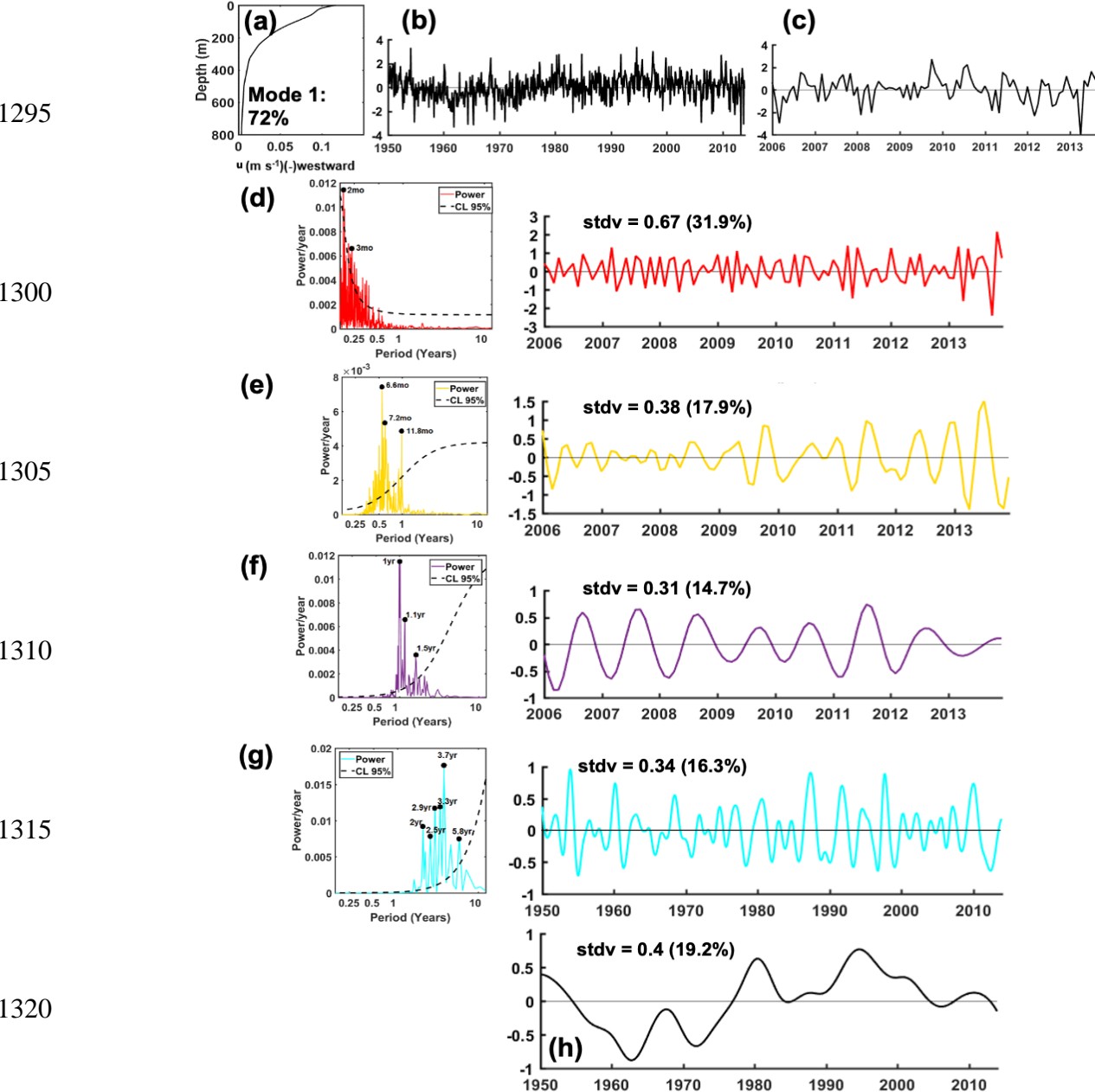

**Figure 12. Same as in Fig. 10, except for the point C$_{EJ}$ with the temporal variability of EOF1 accounting for 72% of total variance.**

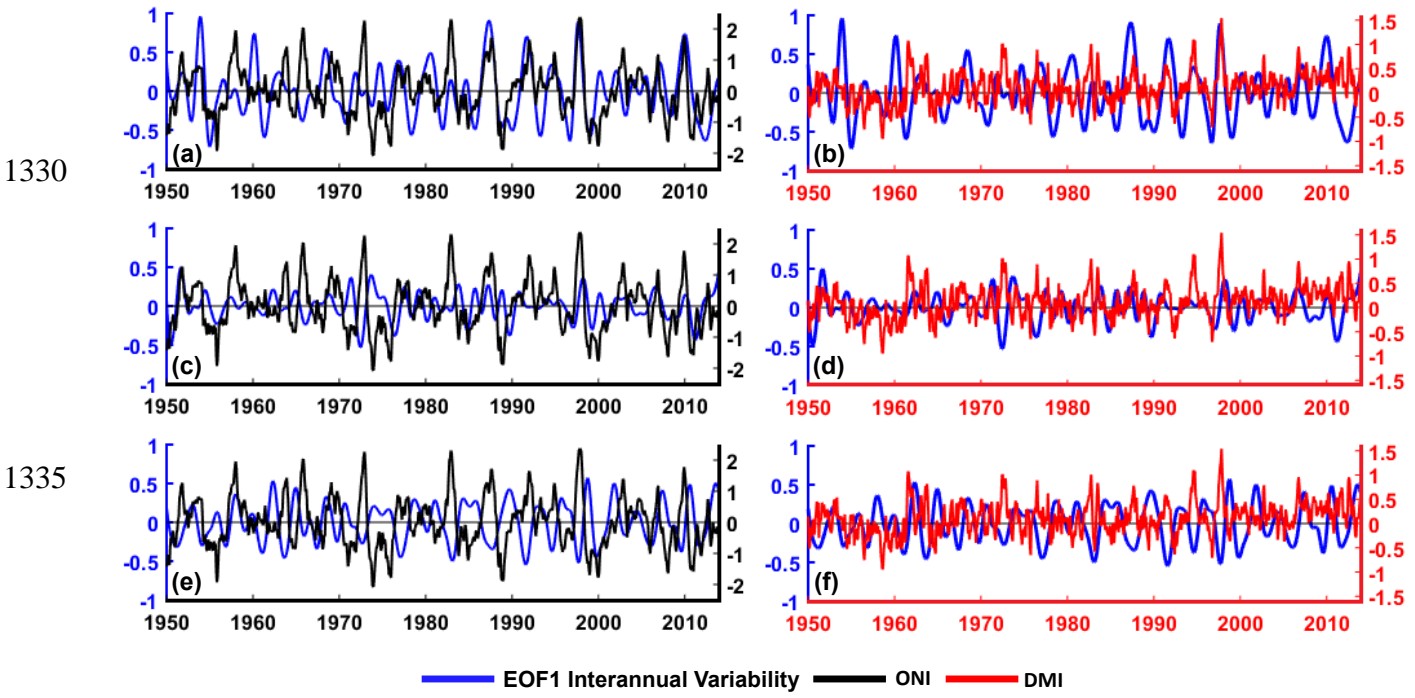



Figure 13. The interannual variability of the EOF first temporal mode (blue lines) is overlaid with ONI (black lines) and DMI (red lines) at $C_{EJ}$ (a and b), $A_{WJ}$ (c and d), and $B_{SM}$ (e and f).







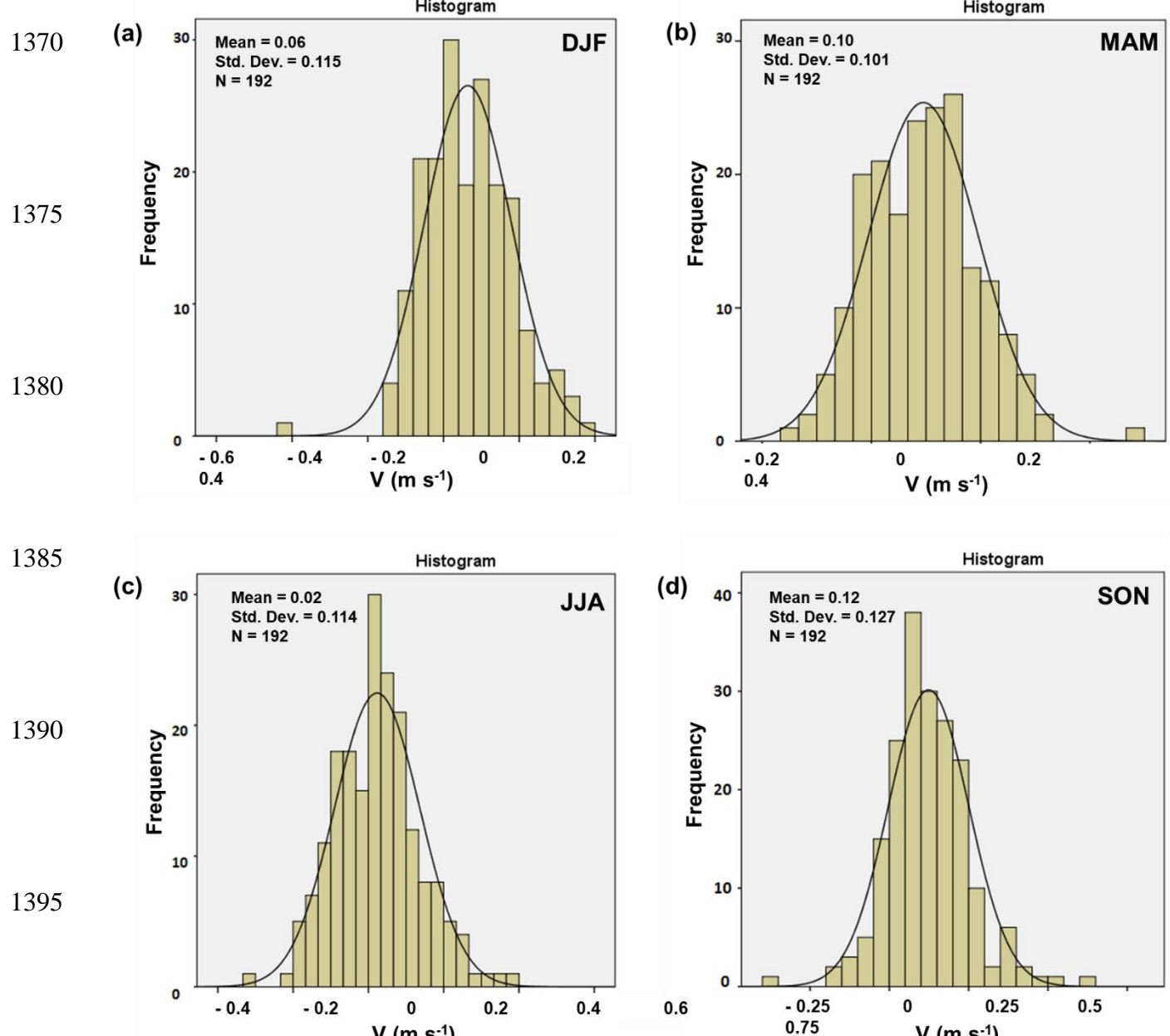

**Figure 14. A probability distribution function of the EOF1 of zonal currents for each of the NW (a), SE (c), and transition (b and d) seasons at B$_{SM}$ at a depth of ~40 m.**

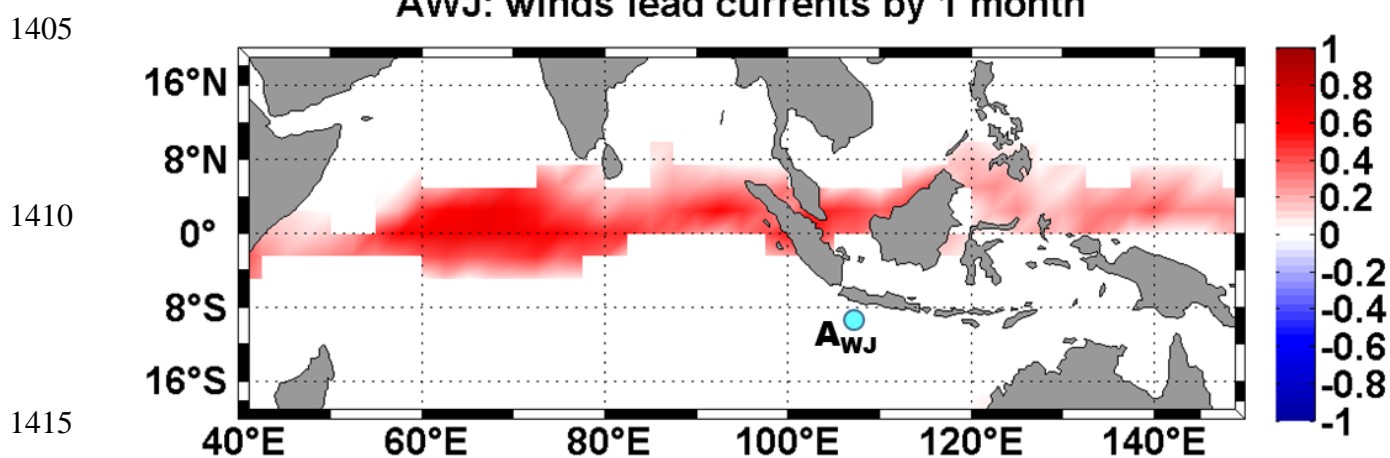

Figure 15. A correlation map between zonal wind and zonal currents (at 30 m) at A$_{WJ}$ for the semiannual signals extracted using the EEMD method. The 95% significance level is approximately ± 0.07.

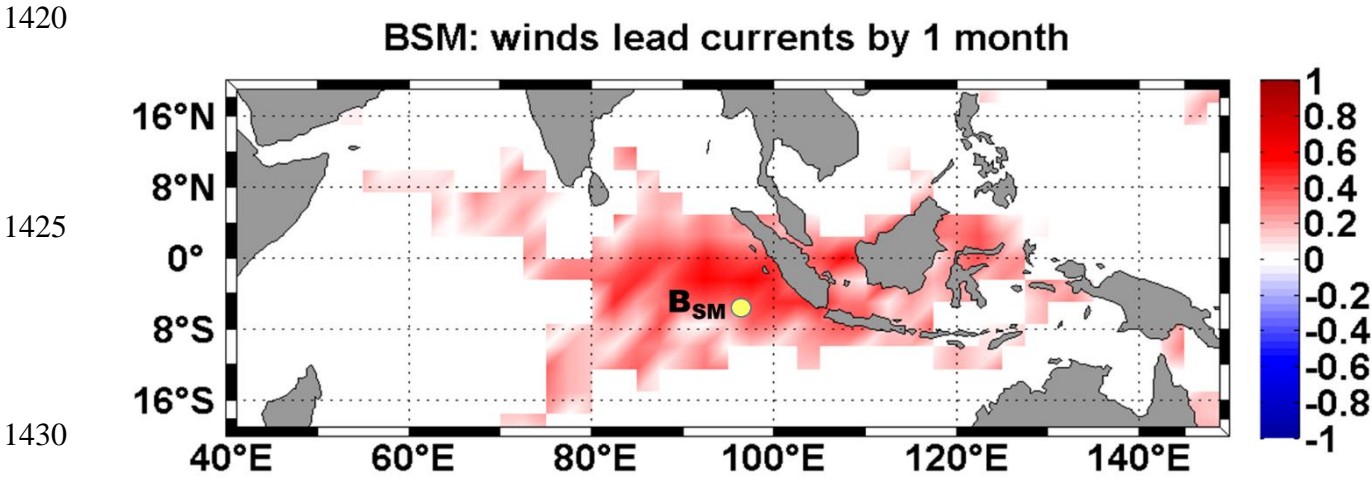

Figure 16. As in Fig. 15, but at B$_{SM}$ and for interannual signal.

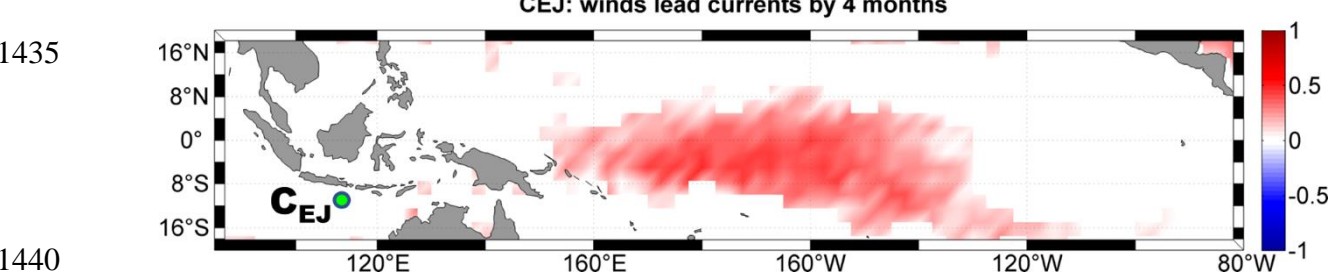

Figure 17. As in Fig. 15, but at C$_{EJ}$ and for interannual signal.