# Peer review of "Simulated Zonal Current Characteristics in the Southeastern Tropical Indian Ocean (SETIO)"

_Ocean Science, 2020_

## Referee Comment (RC1) · Anonymous Referee #1 · 3 Dec 2020

This study investigate the variability of zonal current in the Southeastern Tropical Indian Ocean using HYCOM simulations. The authors described the simulated features of zonal currents in the SJC and ITF/SEC region and examined the intraseasonal to interannual variability of zonal currents in this region. Some interesting results are reported, but a major revision is needed. The major problem with this manuscript is the methodology the authors used. Please see my following comments for details.

Major comments: 1. The authors described the simulated "Vertical structure of zonal current" along transects A, B and C in sections 3.2.1, 3.2.2 and 3.2.3. However, it does not make sense to discuss the "zonal current" along a southeast-northwest section like transects A and B, and especially when there is no clear mean currents along the transects. The meridional components of velocity in the transects A and B are

obviously important as shown in Figures 1a and 2. I suggest that the authors use meridional sections (e.g., AEJ-BEJ-CEJ).

2. The authors used EOF analysis to the zonal velocity at selected points, e.g., AWJ, to investigate the variability of zonal currents SJC, ITF and SEC. But unfortunately, the zonal component of velocity at a selected point is obviously not the currents they aimed to study, just considering that the zonal currents are not steady and usually swing horizontally and that the meridional components of the velocity are non-negligible for these coastal currents. As I said, it might be good to examine the EOF modes of zonal currents across the meridional sections.

3. The authors used EEMD analysis to the EOF1 (PC1) as well, but I do not understand why the authors did this. The EOF1 itself represents a mode, which means an eigenmode with an eigenperiod. Then, why an eigenmode could be further decomposed into various modes with various periods?

Minor comments: This study investigates the variability of zonal current in the Southeastern Tropical Indian Ocean using HYCOM simulations. Hence, the characteristics described in the manuscript might be depended on the model. The authors may moderately change the title, for example, "Simulated Zonal Current Characteristics in the Southeastern Tropical Indian Ocean (SETIO)"

Line 40-45ïïjŽ I do not understand the logical relationship here between previous studies and what you said after "Hence". What is the scientific question that is not understood in previous studies and what is your purpose? That should specified clearly and unambiguously.

Line 47: "have been carried out by previous investigators" -> have been investigated

Lines 65-70: The authors may also review the salinity effect in the inter-annual and decadal variability of ITF. For example, Hu and Sprintall 2016, JGR; 2017, GRL; Jyoti et al., 2019. The salinity effect mechanism is an important component of ITF dynamics

different from the wind forcing mechanism.

Line 100: Does the HYCOM assimilate surface observations? Lines 132-133: No necessary to repeat the references of EEMD here Section 3.1: A longitude-depth plot of mean zonal currents along the three sections should be presented.

Please also note the supplement to this comment:
https://os.copernicus.org/preprints/os-2020-91/os-2020-91-RC1-supplement.pdf

---

## Referee Comment (RC2) · Anonymous Referee #2 · 10 Dec 2020

This study investigates the variability of zonal current within the region that involves multiple currents system, i.e., the South Java Current, the Indonesian Throughflow, the South Equatorial Current, and the coastal upwelling. Due to interactions among these currents, the variability in this region is very complex. Using EOF and EEMD, the authors tried to discuss the intraseasonal, seasonal, and interannual variations of the zonal currents in the southeastern tropical Indian Ocean. Althrough the results and text are generally easy to follow, the analysis of the paper, however, is mainly based on correlation analysis, the dynamical explanation is too weak. Moreover, this paper does not give an improved view of the currents variations in the southeastern tropical Indian Ocean. Thereby, I could not recommend the present manuscript being accepted.

---

## Author Comment (AC1) · 15 Jan 2021

Thank you for your time and effort in reviewing our paper, and for providing the valuable suggestions, comments, and corrections, which helped make our manuscript stronger. We have modified the manuscript based on the Reviewer's suggestions and hope that the revision adequately addressed the Reviewer's concerns, so that the revised manuscript will be suitable for publication.

Anonymous Referee #1

This study investigates the variability of zonal current in the Southeastern Tropical Indian Ocean using HYCOM simulations. The authors described the simulated features

of zonal currents in the SJC and ITF/SEC region and examined the intraseasonal to interannual variability of zonal currents in this region. Some interesting results are reported, but a major revision is needed. The major problem with this manuscript is the methodology the authors used. Please see my following comments for details.

Major comments:

1. The authors described the simulated "Vertical structure of zonal current" along transects A, B and C in sections 3.2.1, 3.2.2 and 3.2.3. However, it does not make sense to discuss the "zonal current" along a southeast-northwest section like transects A and B, and especially when there is no clear mean currents along the transects. The meridional components of velocity in the transects A and B are obviously important as shown in Figures 1a and 2. I suggest that the authors use meridional sections (e.g., AEJ-BEJ-CEJ).

Thank you for your suggestions. We have revised the description of "Vertical structure of zonal current" by using meridional sections (East Java: AEJ-BEJ-CEJ; West Java: AWJ-BWJ-CWJ; and Sumatra: ASM-BSM-CSM). In addition, the revision can also be seen at the end of this response for details.

2. The authors used EOF analysis to the zonal velocity at selected points, e.g., AWJ, to investigate the variability of zonal currents SJC, ITF and SEC. But unfortunately, the zonal component of velocity at a selected point is obviously not the currents they aimed to study, just considering that the zonal currents are not steady and usually swing horizontally and that the meridional components of the velocity are non-negligible for these coastal currents. As I said, it might be good to examine the EOF modes of zonal currents across the meridional sections.

Thank you for your valuable suggestions. We have added examination of the EOF modes of zonal currents across the meridional sections, namely sections East Java (EJ), West Java (WJ), and Sumatra (SM), as follows:

[revised manuscript text omitted]

3. The authors used EEMD analysis to the EOF1 (PC1) as well, but I do not understand why the authors did this. The EOF1 itself represents a mode, which means an eigenmode with an eigenperiod. Then, why an eigenmode could be further decomposed into various modes with various periods?

Thank you for the comments. We would like to clarify it.

EOF via singular value decomposition matrix of data decomposes into three new matrices which may be used to form the eigenvalues, eigenvectors, and eigenfunctions. The resulting eigenfunctions display the spatial patterns (in our case, the vertical patterns for mode-1 shown in Figs. 5a, 6a, 7a), while the eigenvectors (eigenperiod) show time varying amplitude for each mode (in our case, the eigenperiod for mode-1 shown in Figs. 5b, 6b, and 7b). The eigenvalues demonstrate the strength of each mode, with the first mode defines as being the mode associated with the largest eigenvalue, and hence the largest percent of the variance (Table 1).

To find out what frequencies are dominant in the eigenperiod of mode-1, one can apply Fourier transform of these time series (Figs. 5b, 6b, and 7b) by assuming that the time varying signal is linear and periodic. In here, we use time series analysis technique (EEMD) which is using Hilbert transform, that suitable not only for linear periodic time series, but also suitable for nonlinear and nonperiodic signals. For detailed technique, please refer to Huang et al., 1998. Our results are shown in Figs. 5d-h, 6d-h, and 7d-h.

We do hope this could clarify it. Thank you

Minor comments:

1. This study investigates the variability of zonal current in the Southeastern Tropical Indian Ocean using HYCOM simulations. Hence, the characteristics described in the manuscript might be depended on the model. The authors may moderately change the title, for example, "Simulated Zonal Current Characteristics in the Southeastern Tropical Indian Ocean (SETIO)".

Thank you for your suggestions. As suggested by the reviewer, we have changed the title to "Simulated Zonal Current Characteristics in the Southeastern Tropical Indian Ocean (SETIO)".

2. Line 40-45: I do not understand the logical relationship here between previous studies and what you said after "Hence". What is the scientific question that is not understood in previous studies and what is your purpose? That should specified clearly

and unambiguously.

Thank you very much for the careful reading of our manuscript. To make it more obvious and easier to understand the narration, we have reorganized the description, as follow:

Regarding dynamics and characteristics of the SETIO, especially adjacent to the western coast of Sumatra and the southern coast of Java, all previous investigations are either based on numerical model, remote sensed data or velocity/moorings observations within the Indonesian seas or at the exit passages of Indonesian seas (Sunda, Lombok, Ombai, and Timor passages) which lead into the SETIO. There is almost no ocean current/velocity measurement within the SETIO. The observational velocity data are available only at limited points in space and time. The only velocity measurement in south of Java or in the SETIO region reported by Sprintall et al. (1999). The mooring was deployed south of Java in 200 m water depth from March 1997 to March 1998 at depths of 55 m, 115 m and 175 m velocity measurements, but only current meters at 115 m and 175 m were fully working properly (Sprintall et al., 1999). Recently, there are some moorings to measure velocity and stratification deployed in the SETIO region. However, they have not been fully recovered nor published. Hence, due to limited in situ velocity measurements in the SETIO, the detailed dynamics and characteristics of ocean currents in the region have not been fully explained. It is important to obtain a better understanding of current characteristics as well as their spatial and temporal variations in the SETIO adjacent to the southern coasts of Sumatra and Java both for scientific and practical reasons, such as fisheries, climate, and navigation. These are the main motivations of the present study.

3. Line 47: "have been carried out by previous investigators" -> have been investigated

Thank you very much for your correction. We have changed "have been carried out" to "have been investigated".

4. Lines 65-70: The authors may also review the salinity effect in the inter-annual and

decadal variability of ITF. For example, Hu and Sprintall 2016, JGR; 2017, GRL; Jyoti et al., 2019. The salinity effect mechanism is an important component of ITF dynamics different from the wind forcing mechanism.

Thank you very much for your suggestion. We have added the review of the salinity effect in the inter-annual and decadal variability of ITF as suggested by the reviewer, as follow:

In addition to the wind forcing mechanism, fluctuations in rainfall over the Indonesian Seas that modulates salinity also influences the ITF transport on interannual (Hu and Sprintall, 2016) and decadal (Hu and Sprintall, 2017; Jyoti et al., 2019) time scales. They found that the salinity effect mechanism is an important component of ITF dynamics and it is different from the wind forcing mechanism. Moreover, it has been revealed that salinity effect contributes 36% of the total interannual variability of the ITF transport (Hu and Sprintall, 2016) and dominates an increasing trend of the ITF transport during the past decade (Hu and Sprintall, 2017).

5. Line 100: Does the HYCOM assimilate surface observations?

Yes, the HYCOM assimilates surface skin temperatures from NCEP Reanalysis Data (4 times daily).

6. Lines 132-133: No necessary to repeat the references of EEMD here

Thank you very much for your correction. We have deleted the references of EEMD.

7. Section 3.1: A longitude-depth plot of mean zonal currents along the three sections should be presented.

Thank you very much for your suggestion. We have added a longitude-depth plot of mean zonal currents along the three sections (Fig. 8) and additional description of it, as follow:

Here, longitude-depth plots of mean zonal currents along the sections A, B, and C
are also presented in Fig. 8, which clearly shows different zonal current system along the transects. Mean zonal currents along Transect A (Fig. 8a) show two distinctive features: (1) the mean currents dominantly flow eastward from the sea surface to 100 m depth (95° E–114° E), and (2) they are predominantly westward from the region (115° E), which is close to Lombok Strait (LS) as one of the ITF exit passages, to the 122° E longitude line. In addition, the mean eastward current at AEJ occurs up to ∼600 m. Meanwhile, the average current on Transect B (the transitional zone) is westward, especially at longitudes 101° E to 107° E (Fig. 8b). In the offshore region (Transect C), mean zonal current flows westward throughout the region (Fig. 8c).

= = = = = = = = = = = = = = = = =

Revision of Major comments # 1: description revision of "Vertical structure of zonal current" in sections 3.2.1, 3.2.2 and 3.2.3 by using meridional sections.

[revised manuscript text omitted]
 AWJ (Figs. 3g and 3j) is weaker than that at AEJ (Figs. 3a and 3d). The weaker current at AWJ may exist as a consequence of the weaker mean NW monsoon at this point compared with that at AEJ (Fig. 9). Interestingly, at a depth of 100 m, there is a maximum westward current at AWJ during DJF with velocity of about 0.1 m s-1 (Figs. 3g and 3j). Here, we suggest that ITF is the cause of the westward current at 100 m at AWJ during the DJF. In regard to the ITF, Fig. 3 of Sprintall et al. (2010) shows cores of subsurface maximum ITF extending from 100 m to 250 m depth in the northern part of the Ombai Strait and from 100 m to 800 m depth at the southern part of the strait during DJF. Meanwhile, the influence of ITF on the zonal current at AEJ at 100 m is weaker as a consequence of the stronger NW monsoon at AEJ compared with those at AWJ (Fig. 9), so that the current flows rather eastward at AEJ during DJF (Figs. 3a and 3d).

To further investigate which one is more influential between the ITF and the NW monsoon to force the zonal current at the AWJ and AEJ at 100 m depth, we have carried out correlation between the zonal current at both points (each at depth of ~100 m) and each the NW zonal wind and the zonal current representing subsur-

face (~200 m) maximum ITF in the southern Ombai Strait (Table 2 in Supplementary File). Here, the ITF in the southern part of the Ombai Strait was chosen for carrying out the correlation because the ITF flows mainly through the southern part of the passage (Sprintall et al., 2010). It is found that the subsurface maximum ITF during DJF exists at a depth of about 200 m in both the northern and southern parts of the Ombai Strait and it is stronger during DJF than JJA in both parts of the strait (Fig. 3 of Sprintall et al., 2010). In this study, the DJF zonal currents in the period of 2004 through 2006 in the southern Ombai Strait derived from the INSTANT program (http://www.marine.csiro.au/~cow074/ instantdata.htm) were used for the correlation analysis.

It is found that during DJF the zonal current at AWJ at 100 m shows high correlation with the subsurface (~200 m) maximum ITF in the southern Ombai Strait, whereas its correlation with the NW zonal wind is weak (Table 2). Moreover, although during DJF the correlations between the zonal current at AEJ at 100 m and each NW zonal wind and subsurface (~200 m) maximum ITF in the southern Ombai Strait are below the significance level, the NW zonal wind is more influential to force variation of zonal current at AEJ at 100 m than the ITF. Hence, during DJF we suggest that the westward current simulated at AWJ at 100 m is ITF-related, whereas that at AEJ is relatively NW zonal wind-related. As already discussed, in addition to the local eastward winds during DJF, it is suggested that the arrival of downwelling Kelvin waves in December/January at AEJ may contribute to a net eastward current across the water column, which in turn reducing the influence of ITF at this point.

In the transition region, the mean current at BWJ is westward and it is more dominated by the ITF (denoted by black lines in Figs. 3g-l). Similar to BEJ, the seasonal feature of the subsurface maximum ITF is also found at BWJ in which the corresponding westward currents at this point reaches its maximum value at ~100 m depth and it is stronger during DJF than JJA and (Figs. 3h and 3k). In this study, it is also found that the zonal westward currents at 100 m depth at BWJ has a strong correlation with

the subsurface (∼200 m) maximum ITF in the southern Ombai Strait, with correlation coefficient about 0.77 and with the 95% significance level approximately ±0.33, corroborating that the ITF flowing from the Ombai Strait is the main driver for zonal westward current at this point.

Furthermore, like CEJ, characteristic of persistent westward current exists in the offshore region (CWJ), which is attributed to SEC and the westward current has mean velocity around 0.22–0.33 m s-1 in the upper 100 m (Figs. 3i and 3l). The simulated westward current at CWJ shows seasonal variations and reaches its maximum value of about 0.48 m s-1.

3.2.3 Vertical Structure of Zonal Current along Meridional Section Sumatra (ASM-BSM-CSM)

Vertical structures of zonal current along the meridional transect Sumatra (SM; ASM-BSM-CSM) are shown in Figs. 3m-r. Similar to AEJ and AWJ, mean zonal current at ASM (nearshore region) is eastward, attributed to SJC, and associated with the Kelvin wave propagation. However, due to ASM located in front of west of Sumatra Island (Fig. 2), which is oriented in the northwest-southeast direction, the meridional components of velocity at this point is also dominant (Figs. 1a and 2). Therefore, zonal currents at ASM are relatively weaker than those at AWJ and AEJ, which are located in front of south of Java Island oriented in the west-east direction. For example, during SON, the eastward current reaches its maximum velocity of about 0.05 m s-1 at ASM (cyan lines in Figs. 3m and 3p), whereas it is about 0.23 m s-1 (at AWJ; Figs. 3g and 3j) and 0.20 m s-1 (at AEJ; Figs. 3a and 3d) at ∼30-50 m depths.

Furthermore, results of this study show that a maximum value of the eastward current forced by a Kelvin wave at ASM, AWJ, and AEJ is found at a certain depth (at ∼30-50 m depths) and it is supposed to be attributed to a baroclinic Kelvin wave. The baroclinic Kelvin wave propagating vertically and horizontally along its waveguide can exert energy the most at a certain depth (Drushka et al., 2010; Pujiana et al., 2013; Iskandar

et al., 2014). According to laboratory experiment observation conducted by Codiga et al. (1999) and Hallock et al. (2009), Kelvin wave can be trapped in a slope and propagates along an isobath. This phenomenon is known as slope-trapped baroclinic Kelvin wave. Moreover, Kelvin wave which propagates along continental slope with strong stratification can cause strong current velocity. Codiga et al. (1999) also found that this slope Kelvin wave is formed after encountering a canyon-like bathymetry. Meanwhile, Pujiana et al. (2013) shows that Kelvin wave propagation from Lombok Strait to Makassar Strait, across Sunda continental slope, is along isobaths at depths greater than 50 m. In this present study, eastward current along the Transect A has maximum current velocity at depth ∼30–50 m. Therefore, it is suggested that this maximum eastward current at ∼30–50 m depth associated with slope-trapped Kelvin wave, which propagates at that depth along the southern coasts of Sumatra and Java.

In the transition region, characteristic of average zonal current (the climatological current field) at BSM (Figs. 3n and 3q) is different from that at BWJ (Figs. 3h and 3k) and BEJ (Figs. 3b and 3e). The average current at BSM is eastward, while at points BWJ and BEJ it is westward. During NW and transitional periods of the monsoon, zonal current at BSM flows eastward and reaches its maximum velocity of about 0.12 m s-1 at a depth of 40 m within the period of SON (Fig. 3q). Meanwhile, during SE monsoon, the zonal current at this point flows westward. In contrast to the mean zonal currents in the nearshore region (ASM), it seems that the average zonal current field at BSM is not attributed to SJC. The reason is the BSM location, which is far from the coasts of Mentawai Islands and Enggano Island off the western coast of Sumatra by 430 km. This distance is more than Rossby radius of deformation at this latitude (∼90 km). Thereby, Kelvin waves, which affect the SJC variations, do not exist at this point. We suggest that the current variability at BSM is influenced by tropical current systems in the Indian Ocean, such as the Equatorial Counter Current (ECC), Southwest Monsoon Current (SWMC), and Wyrtki Jet. Here, we displayed seasonal averaged surface currents over 64 years (1950–2013) and schematics of the tropical current systems in the Indian Ocean as supporting evidence (Fig. 10).

Figure 10 shows that BSM is located at an area, which is affected by the ECC, SWMC, and Wyrtki Jet. It can be seen in the Fig. 10a that during DJF, surface currents along the equatorial Indian Ocean is dominated by the westward North Equatorial Current (NEC) and the eastward ECC. Meanwhile, during JJA (Fig. 10c), the NEC disappears and the ECC becomes absorbed into the SWMC, which dominantly flows eastward in the northern Indian Ocean (Tomczak and Godfrey, 1994). In addition, during the transitional periods (MAM and SON), the Jet is generated, and it causes a strengthening of eastward flows along the equatorial Indian Ocean (Figs. 10b and 10d). This explains the cause of climatological current at BSM flows eastward and reaches its maximum velocity during SON and MAM. These currents (the ECC, SWMC, and Wyrtki Jet) flow eastward before they turn and some part of their flow feed into the SEC in the southern Indian Ocean.

Current characteristics in the offshore region (CSM) generally show similarities with those at CWJ and CEJ, as shown in Figs. 3o and 3r. The current at CSM is attributed to SEC and flows westward all year round, with mean velocity around 0.18–0.3 m s-1 in the upper 100 m. In addition, the strength of westward current at CSM varies seasonally and reaches its maximum value of about 0.42 m s-1 during SON (Fig. 3r).

Please also note the supplement to this comment:
https://os.copernicus.org/preprints/os-2020-91/os-2020-91-AC1-supplement.pdf
* * *
[Figure]

**Fig. 1.** Validation of HYCOM zonal currents with OSCAR and RAMA datasets: (a) Locations of validation points: Points O1 (8oS, 116oE), O2 (7oS, 98oE), and O3 (11.5oS, 113oE) for the OSCAR data, while R1 (0oS, 9

[Figure]

**Fig. 2.** The area of study interest in the SETIO region adjacent to the Sumatra-Java southern coasts. The blue arrows show climatological (yearly mean) surface (1 m) current field over 64 years from 1950 to 20

[Figure]

**Fig. 3.** Mean and seasonal depth profiles of zonal current velocity derived from the HYCOM simulation results for the period of 1950 through 2013, at points: (a) AEJ, (b) BEJ, (c) CEJ, (g) AWJ, (h) BWJ, (i) CW

[Figure]

**Fig. 4.** Vertical structures (a, c, and e) and their associated temporal variability of EOF1 (b, d, and f) of zonal currents along the three meridional sections: EJ (a and b), WJ (c and d), and SM (e and f). I

[Figure]

**Fig. 5.** (a) Vertical structure and (b) its associated temporal variability of EOF1 (58% of total variance) at the point AWJ. (c) As (b), except for the last eight-year period of the EOF1. The EEMD is then ap

[Figure]

**Fig. 6.** Same as in Figure 5, except for the point BSM with the temporal variability of EOF1 accounting for 64% of total variance.

[Figure]

**Fig. 7.** Same as in Figure 5, except for the point CEJ with the temporal variability of EOF1 accounting for 72% of total variance.

[Figure]

**Fig. 8.** Longitude-depth profiles of mean zonal currents along (a) Transect A, (b) Transect B, and (c) Transect C. Positive (negative) values of the zonal currents indicate eastward (westward). Green dash line

[Figure]

**Fig. 9.** Mean NW monsoon for the period of 1950 to 2013 (climatological wind field during the DJF).

[Figure]

**Fig. 10.** Seasonal averaged surface (1 m) currents over 64 years (1950-2013) and schematics of the tropical current systems in the Indian Ocean during (a) DJF, (b) MAM, (c) JJA, and (d) SON. Current branches in

**Supplement:**

**List of Changes and Responses for Ocean Science Discussion (OS-2020-91)**

Interactive comment on
"Zonal Current Characteristics in the Southeastern Tropical Indian Ocean (SETIO)"

Thank you for your time and effort in reviewing our paper, and for providing the valuable suggestions, comments, and corrections, which helped make our manuscript stronger. We have modified the manuscript based on the Reviewer's suggestions and hope that the revision adequately addressed the Reviewer's concerns, so that the revised manuscript will be suitable for publication.

Anonymous Referee #1

This study investigates the variability of zonal current in the Southeastern Tropical Indian Ocean using HYCOM simulations. The authors described the simulated features of zonal currents in the SJC and ITF/SEC region and examined the intraseasonal to interannual variability of zonal currents in this region. Some interesting results are reported, but a major revision is needed. The major problem with this manuscript is the methodology the authors used. Please see my following comments for details.

**Major comments:**
1.  The authors described the simulated "Vertical structure of zonal current" along transects A, B and C in sections 3.2.1, 3.2.2 and 3.2.3. However, it does not make sense to discuss the "zonal current" along a southeast-northwest section like transects A and B, and especially when there is no clear mean currents along the transects. The meridional components of velocity in the transects A and B are obviously important as shown in Figures 1a and 2. I suggest that the authors use meridional sections (e.g., AEJ-BEJ-CEJ).

Thank you for your suggestions. We have revised the description of "Vertical structure of zonal current" by using meridional sections (East Java: AEJ-BEJ-CEJ; West Java: AWJ-BWJ-CWJ; and Sumatra: ASM-BSM-CSM). In addition, the revision can also be seen at the end of this response for details.

[Figure]

Figure 1. Validation of HYCOM zonal currents with OSCAR and RAMA datasets: (a) Locations of validation points: Points O1 (8ºS, 116ºE), O2 (7ºS, 98ºE), and O3 (11.5ºS, 113ºE) for the OSCAR data, while R1 (0ºS, 90ºE) and R2 (8.5ºS, 106.75ºE) for the RAMA data. (b)-(d) Time series of the zonal currents observed by the HYCOM (blue lines) and the OSCAR (red lines) at a depth of 0.5 m at point O1, O2, and O3, respectively. Meanwhile (e)-(h) are the time series of zonal currents observed by the HYCOM (blue lines) and the moored RAMA (red lines) at point R1 at depths of 50, 150, 250, and 350 m, sequentially. Meanwhile, (i)-(k) are the same as (e)-(h), except for point R2 and depths of 40, 80, and 120 m, respectively. In the Figures 1e-h (point R1), a monthly low-pass filter has been applied before plotting.

[Figure]

Figure 2. The area of study interest in the SETIO region adjacent to the Sumatra-Java southern coasts. The blue arrows show climatological (yearly mean) surface (1 m) current field over 64 years from 1950 to 2013. Yellow lines are the meridional sections along the three longitudes (98°E, 107°E, and 113°E), while red lines are the three selected transects: A, B, and C. Green, yellow, and cyan circles are the locations in which the zonal currents are analysed, namely points ASM, AWJ, AEJ (on the Transect A); points BSM, BWJ, and BEJ (on the Transect B); and points CSM, CWJ, and CEJ (on the Transect C). The subscripts SM, WJ, and EJ denote regions which close to Sumatra, West Java, and East Java.

2. The authors used EOF analysis to the zonal velocity at selected points, e.g., AWJ, to investigate the variability of zonal currents SJC, ITF and SEC. But unfortunately, the zonal component of velocity at a selected point is obviously not the currents they aimed to study, just considering that the zonal currents are not steady and usually swing horizontally and that the meridional components of the velocity are non-negligible for these coastal currents.

As I said, it might be good to examine the EOF modes of zonal currents across the meridional sections.

Thank you for your valuable suggestions. We have added examination of the EOF modes of zonal currents across the meridional sections, namely sections East Java (EJ), West Java (WJ), and Sumatra (SM), as follows:

EOF analysis gives vertical mode structures (spatial mode) and their normalized temporal mode variabilities relative to the mean which influence zonal current variability in the study area. Before performing the EOF analysis, the average value of the current data has been removed (solid black lines in the Figs. 3a-r). To further analyze the zonal current characteristics in the nearshore and offshore areas, and the transition region between them, we examined the EOF modes of zonal current across the three meridional sections (EJ, WJ, and SM). In this paper, we only considered the first mode of EOF (EOF1) analysis since it is associated with the largest percent of the variance. Figure 4 shows vertical structures and their associated temporal variability of EOF1 of zonal currents along the meridional sections. Here, as an example, the temporal variability is only shown for the last eight-year period of the EOF1 (2006 to 2013). It can be clearly seen that remarkable features of zonal currents are revealed between nearshore and offshore areas as well as in the three meridional sections (Fig. 4).

[Figure]

Figure 3. Mean and seasonal depth profiles of zonal current velocity derived from the HYCOM simulation results for the period of 1950 through 2013, at points: (a) AEJ, (b) BEJ, (c) CEJ, (g) AWJ, (h) BWJ, (i) CWJ, (m) ASM, (n) BSM, and (o) CSM. Meanwhile, (d)-(f), (j)-(l), and (p)-(r) are the same as (a)-(c), (g)-(i), and (m)-(o), respectively, except for depths of 0-100 m.

[revised manuscript text omitted]

3. The authors used EEMD analysis to the EOF1 (PC1) as well, but I do not understand why the authors did this. The EOF1 itself represents a mode, which means an eigen-mode with an eigenperiod. Then, why an eigenmode could be further decomposed into various modes with various periods?

Thank you for the comments. We would like to clarify it.

EOF via singular value decomposition matrix of data decomposes into three new matrices which may be used to form the eigenvalues, eigenvectors, and eigenfunctions. The resulting eigenfunctions display the spatial patterns (in our case, the vertical patterns for mode-1 shown in Figs. 5a, 6a, 7a), while the eigenvectors (eigenperiod) show time varying amplitude for each mode (in our case, the eigenperiod for mode-1 shown in Figs. 5b, 6b, and 7b). The eigenvalues demonstrate the strength of each mode, with the first mode defines as being the mode associated with the largest eigenvalue, and hence the largest percent of the variance (Table 1).

To find out what frequencies are dominant in the eigenperiod of mode-1, one can apply Fourier transform of these time series (Figs. 5b, 6b, and 7b) by assuming that the time varying signal is linear and periodic. In here, we use time series analysis technique (EEMD) which is using Hilbert transform, that suitable not only for linear periodic time series, but also suitable for nonlinear and nonperiodic signals. For detailed technique, please refer to Huang et al., 1998. Our results are shown in Figs. 5d-h, 6d-h, and 7d-h.

We do hope this could clarify it. Thank you.

[Figure]

Figure 5. (a) Vertical structure and (b) its associated temporal variability of EOF1 (58% of total variance) at the point A$_{WJ}$. (c) As (b), except for the last eight-year period of the EOF1. The EEMD is then applied to the EOF temporal structure to decompose temporal variability: (d) intraseasonal, (e) semiannual (f) annual, and (g) interannual variabilities with their corresponding red spectrum as a reference for 95% confidence limit (left panel), whereas (h) represents the long-term trend.

[Figure]

Figure 6. Same as in Figure 5, except for the point $B_{SM}$ with the temporal variability of EOF1 accounting for 64% of total variance.

[Figure]

Figure 7. Same as in Figure 5, except for the point $C_{EJ}$ with the temporal variability of EOF1 accounting for 72% of total variance.

**Minor comments:**
1. This study investigates the variability of zonal current in the Southeastern Tropical Indian Ocean using HYCOM simulations. Hence, the characteristics described in the manuscript might be depended on the model. The authors may moderately change the title, for example, "Simulated Zonal Current Characteristics in the Southeastern Tropical Indian Ocean (SETIO)".

Thank you for your suggestions. As suggested by the reviewer, we have changed the title to "Simulated Zonal Current Characteristics in the Southeastern Tropical Indian Ocean (SETIO)".

2. Line 40-45: I do not understand the logical relationship here between previous studies and what you said after "Hence". What is the scientific question that is not understood in previous studies and what is your purpose? That should specified clearly and unambiguously.

Thank you very much for the careful reading of our manuscript. To make it more obvious and easier to understand the narration, we have reorganized the description, as follow:

*Regarding dynamics and characteristics of the SETIO, especially adjacent to the western coast of Sumatra and the southern coast of Java, all previous investigations are either based on numerical model, remote sensed data or velocity/moorings observations within the Indonesian seas or at the exit passages of Indonesian seas (Sunda, Lombok, Ombai, and Timor passages) which lead into the SETIO. There is almost no ocean current/velocity measurement within the SETIO. The observational velocity data are available only at limited points in space and time. The only velocity measurement in south of Java or in the SETIO region reported by Sprintall et al. (1999). The mooring was deployed south of Java in 200 m water depth from March 1997 to March 1998 at depths of 55 m, 115 m and 175 m velocity measurements, but only current meters at 115 m and 175 m were fully working properly (Sprintall et al., 1999). Recently, there are some moorings to measure velocity and stratification deployed in the SETIO region. However, they have not been fully recovered nor published. Hence, due to limited in situ velocity measurements in the SETIO, the detailed dynamics and characteristics of ocean currents in the region have not been fully explained. It is important to obtain a better understanding of current characteristics as well as their spatial and temporal variations in the SETIO adjacent to the southern coasts of Sumatra and Java both for scientific and practical reasons, such as fisheries, climate, and navigation. These are the main motivations of the present study.*

3. Line 47: "have been carried out by previous investigators" -> have been investigated

Thank you very much for your correction. We have changed "have been carried out" to "have been investigated".

4. Lines 65-70: The authors may also review the salinity effect in the inter-annual and decadal variability of ITF. For example, Hu and Sprintall 2016, JGR; 2017, GRL; Jyoti et al., 2019. The salinity effect mechanism is an important component of ITF dynamics different from the wind forcing mechanism.

Thank you very much for your suggestion. We have added the review of the salinity effect in the inter-annual and decadal variability of ITF as suggested by the reviewer, as follow:

*In addition to the wind forcing mechanism, fluctuations in rainfall over the Indonesian Seas that modulates salinity also influences the ITF transport on interannual (Hu and Sprintall, 2016) and decadal (Hu and Sprintall, 2017; Jyoti et al., 2019) time scales. They found that the salinity effect mechanism is an important component of ITF dynamics and it is different from the wind forcing mechanism. Moreover, it has been revealed that salinity effect contributes 36% of the total interannual variability of the ITF transport (Hu and Sprintall, 2016) and*

*dominates an increasing trend of the ITF transport during the past decade (Hu and Sprintall, 2017).*

5.  Line 100: Does the HYCOM assimilate surface observations?

Yes, the HYCOM assimilates surface skin temperatures from NCEP Reanalysis Data (4 times daily).

6.  Lines 132-133: No necessary to repeat the references of EEMD here

Thank you very much for your correction. We have deleted the references of EEMD.

7.  Section 3.1: A longitude-depth plot of mean zonal currents along the three sections should be presented.

Thank you very much for your suggestion. We have added a longitude-depth plot of mean zonal currents along the three sections (Fig. 8) and additional description of it, as follow:

[Figure]

Figure 8. Longitude-depth profiles of mean zonal currents along (a) Transect A, (b) Transect B, and (c) Transect C. Positive (negative) values of the zonal currents indicate eastward (westward). Green dash lines denote longitudes of the nine selected points, whereas dark orange dash lines denote longitude of Lombok Strait (LS).

*Here, longitude-depth plots of mean zonal currents along the sections A, B, and C are also presented in Fig. 8, which clearly shows different zonal current system along the transects. Mean zonal currents along Transect A (Fig. 8a) show two distinctive features: (1) the mean currents dominantly flow eastward from the sea surface to 100 m depth (95° E–114° E), and (2) they are predominantly westward from the region (115° E), which is close to Lombok Strait (LS) as one of the ITF exit passages, to the 122° E longitude line. In addition, the mean*

*eastward current at $A_{EJ}$ occurs up to ~600 m. Meanwhile, the average current on Transect B (the transitional zone) is westward, especially at longitudes 101° E to 107° E (Fig. 8b). In the offshore region (Transect C), mean zonal current flows westward throughout the region (Fig. 8c).*

= = = = = = = = = = = = = = = =

**Revision of Major comments # 1:** description revision of "Vertical structure of zonal current" in sections 3.2.1, 3.2.2 and 3.2.3 by using meridional sections.

[revised manuscript text omitted]

---

## Author Comment (AC2) · 15 Jan 2021

This study investigates the variability of zonal current within the region that involves multiple currents system, i.e., the South Java Current, the Indonesian Throughflow, the South Equatorial Current, and the coastal upwelling. Due to interactions among these currents, the variability in this region is very complex. Using EOF and EEMD, the authors tried to discuss the intraseasonal, seasonal, and interannual variations of the zonal currents in the southeastern tropical Indian Ocean. Although the results and text are generally easy to follow, the analysis of the paper, however, is mainly based on

correlation analysis, the dynamical explanation is too weak. Moreover, this paper does not give an improved view of the currents variations in the southeastern tropical Indian Ocean. Thereby, I could not recommend the present manuscript being accepted.

Thank you for your time and effort in reviewing our paper, and for providing the valuable comments and corrections, which helped make our manuscript stronger and better. In this case, we have revised the manuscript based on suggestions/comments of the Anonymous Referee #1 and hope that the revision adequately addressed the Reviewer's concerns, so that the revised manuscript will be suitable for publication.

---

## Author Response (AR2)

**List of Changes and Responses for Ocean Science Discussion (OS-2020-91)**

Interactive comment on
"Simulated Zonal Current Characteristics in the Southeastern Tropical Indian Ocean (SETIO)"

Thank you for your time and effort in reviewing our paper, and for providing the valuable suggestions, comments, and corrections, which helped make our manuscript stronger. We have modified the manuscript based on the Reviewer's suggestions and hope that the revision adequately addressed the Reviewer's concerns, so that the revised manuscript will be suitable for publication.

**Anonymous Referee #1**
**Submitted on 11 Mar 2021**

I am grateful to the authors for their efforts in improving the manuscript. Most of my previous concerns have been addressed. But I have some further comments for your reference. I agree with another anonymous review that this manuscript failed to specify clearly what the novelty of this work is for the scientific community. The authors presented plenty of analysis and estimates based on their simulations, which I think is a good contribution to the community. But unfortunately, it seems that the poor writing and cumbersome structure of the present version make the manuscript a hard read. I suggest the authors further to revise and refine the article carefully. My comments are listed below.

**Major comments:**
1.  Abstract: The authors may clarify clearly what is new relative to previous studies. Remove less important or uninspiring conclusions from the Abstract and add important or novel conclusions to it. For example, the readers may don't care about the power density of each time scale in the Abstract.

We thank the reviewer for the thoughtful suggestions and comments towards improving our manuscript. In the revised manuscript, we have revised the Abstract (Lines 9-37) and refined the article as well. Less important or uninspiring conclusions has been removed from the Abstract and important or novel conclusions has been added to it.

2.  Lines 190-205: I feel confused about this paragraph. The authors claimed that the SJC and SJUC are caused by downwelling Kelvin waves, but you concluded that the SJC and SJUC are induced by local eastward wind as well and called the downwelling Kelvin waves a minor addition. Then, who is the real cause of the currents? Meanwhile, I don't understand how can a non-stationarity wave forms a steady mean flow?

We would like to thank the reviewer for the insightful and careful review of our manuscript. The SJC and SJUC are caused by local eastward wind. During the NW monsoon (DJF), the dominant eastward current at $A_{EJ}$, particularly that at depth beneath 100 m, strengthens and occurs up to ~800 m, coinciding with the arrival of a seasonal downwelling Kelvin wave originating in the equatorial Indian Ocean during the transitional monsoons (SON). To make it more obvious and easier to understand the description, we have modified and refined the paragraph in Lines 213-232, as follows:

**3.2.1 Vertical Structure of Zonal Current along Meridional Section EJ ($A_{EJ}$-$B_{EJ}$-$C_{EJ}$)**

*Different zonal current system along the meridional transect East Java (EJ; $A_{EJ}$-$B_{EJ}$-$C_{EJ}$) can clearly be seen in Figs. 6a-f. On average, for the period 1950 through 2013, zonal climatological current at $A_{EJ}$ (nearshore area) generally flows eastward from the sea surface to 100 m depth (Figs. 6a and 6d) and reaches its maximum value of about 0.16 m s$^{-1}$. It is suggested that the average zonal current at this point is mainly attributed to SJC and it shows seasonal variations. During the SE monsoon (JJA), the strength of climatological eastward SJC at this point in upper 10 m depth reduces (Fig. 6d). Meanwhile, during the NW monsoon (DJF), the current in the upper 10 m (Fig. 6d) flows more eastward in response to the prevailing northwesterly winds (Fig. 7). In general, the mean eastward current at $A_{EJ}$, during DJF was attributed to local winds. Interestingly, during this monsoon period (DJF), the eastward current at $A_{EJ}$, particularly that at depth beneath 100 m, strengthens and occurs up to ~800 m. Other physical processes may account for the enhanced eastward current at this point. The SJC and SJUC, which are seasonally varying currents and predominantly eastward, are defined as the surface current in the upper 150 m and the subsurface current beneath 150 m down to 1000 m, respectively (Iskandar et al., 2006). The eastward-flowing SJC and SJUC are intensified, coinciding with the arrival of a seasonal downwelling Kelvin wave along the south coast of Java (e.g., Sprintall et al., 1999, 2000; Iskandar et al., 2006). Downwelling Kelvin waves originating in the equatorial Indian Ocean during the transitional monsoons propagate along the coasts of western Sumatra and southern Java with phase speeds ranging from 1.5 to 2.9 m s$^{-1}$ (e.g., Sprintall et al., 2000; Syamsudin et al., 2004; Iskandar et al., 2005). These phase speeds indicate that the downwelling Kelvin waves will arrive at $A_{EJ}$ in 21 – 41 days. In this case, downwelling Kelvin waves generated during the monsoon transition period in November may arrive at $A_{EJ}$ in December/January. Therefore, in addition to the local eastward winds, the downwelling Kelvin waves may also contribute to strengthen the eastward currents at $A_{EJ}$ during the NW monsoon, including those at depth beneath 100 m.*

**Minor comments:**

1. Line 44: simulated data -> simulations

Thank you very much for your correction. In the revised manuscript, we have changed "simulated data" to "simulations" (Line 55).

2. Line 50: delete "by previous investigators"

Thank you for your correction. In the revised manuscript, we have deleted "by previous investigators" (Line 62).

3. Line 61: interannual variability of what?

We thank the reviewer for the correction. In the revised manuscript, we have modified "interannual variability" to "interannual variability of ENSO" (Line 71).

4. Line 63: delete "event"

Thank you very much for your correction. In the revised manuscript, we have deleted "event" (Line 73).

5. Line 75: with -> on a

Thank you for your correction. In the revised manuscript, we have changed "with" to "on a" (Line 83).

6. Lines 79-80: The statement is too absolute to say "no information concerning". You may rephrase this sentence, for example: "However, the seasonal and interannual variations of SEC in the SETIO is unclear yet."

Thank you very much for your suggestion. In the revised manuscript, we have rephrased "no information concerning" to "However, the seasonal and interannual variations of SEC in the SETIO are unclear yet" (Line 89-90).

7. Line 82: all previous investigations -> previous studies

Thank you for your suggestion. In the revised manuscript, we have changed "all previous investigations" to "previous studies" (Line 92).

8. Line 185: seasonal depth profiles -> seasonal mean profiles

Thank you very much for your correction. In the revised manuscript, we have changed "seasonal depth profiles" to "seasonal mean profiles" (Line 208).

9. Line 187: at the nine observation points -> at each point

Thank you for your correction. In the revised manuscript, we have changed "at the nine observation points" to "at each point" (Line 210).

10. Line 188 and elsewhere: used too many "distinctive"

Thank you for your valuable suggestions. In the revised manuscript, we have reduced the use of "distinctive" in Line 210 and elsewhere.

11. Figure 4: add unit of longitudes

We thank the reviewer for the suggestions. In the revised manuscript, we have added unit of longitudes (Figure 4).

12. Figures 5 and 9: add unit of latitudes

Thank you for your suggestions. We have added unit of latitudes (Figure 5 and 9) in the revised manuscript

13. Figures 10-12: could you estimate the proportion of contribution of each EEMD mode to the EOF1? For example, standard deviation of each EEMD mode relative to the total variance of PC1?

Thank you for your valuable suggestions. As suggested by the reviewer, we have estimated the proportion of contribution of each EEMD mode to the EOF1 by calculating standard deviation of each EEMD mode relative to the total variance of PC1. In Section 3.3 (Line 387-394), we have added description of the proportion of contribution of each EEMD mode to the EOF1, as follows:

*Moreover, the proportion of contribution of each EEMD mode to the EOF1 is estimated by calculating standard deviation of each EEMD mode relative to the total variance of PC1 (Figs. 10-12). In general, the contributions of each EEMD mode to the EOF1 at $A_{WJ}$ and $B_{SM}$, from largest to smallest, are intraseasonal, semiannual, annual, interannual, and long-term (Figs. 10 and 11). Intriguingly, however, the contribution of long-term signal (19.2 %) at $C_{EJ}$ is larger than the interannual (16.3%) and annual (14.7%) signals (Fig. 12). For the scope of this paper, we only focused on the analysis of the EOF1 of zonal current from intraseasonal to interannual timescales. The interesting results concerning the existence of pronounced contribution of long-term variation to the EOF1 at $C_{EJ}$ will be investigated in a future study.*

**Anonymous Referee #2**

**Submitted on 06 Mar 2021**

This study investigates the variability of zonal current within the region that involves multiple currents system, i.e., the South Java Current, the Indonesian Throughflow, the South Equatorial Current, and the coastal upwelling. Due to interactions among these currents, the variability in this region is very complex. Using EOF and EEMD, the authors have discussed the intraseasonal, seasonal, and interannual variations of the zonal currents in the southeastern tropical Indian Ocean. The authors have improved their manuscript, the results are reasonable and the text are generally easy to follow. I think this study basically meets the goals of the Journal, and thus should be accepted after some major revision, as listed below.

**Major comments:**
1. Although the authors explained why they have done EEMD after EOF, there are some issues need to be further clarified. First, what are the physical mean of the EOF1? In other words, the EOF1 spatial mode is subject to multi-time scale variations, making it confusing that which dynamical processes are dominant.

We thank the reviewer for the thoughtful comments towards improving our manuscript. We would like to clarify it.

EOF via singular value decomposition matrix of data decomposes into three new matrices which may be used to form the eigenvalues, eigenvectors, and eigenfunctions. The resulting eigenfunctions display the spatial patterns (in our case, the vertical patterns for mode-1 shown in Figs. 10a, 11a, 12a), while the eigenvectors (eigenperiod) show time varying amplitude for each mode (in our case, the eigenperiod for mode-1 shown in Figs. 10b, 11b, and 12b). The eigenvalues demonstrate the strength of each mode, with the first mode defines as being the mode associated with the largest eigenvalue, and hence the largest percent of the variance (Table 2).

In our case, the physical mean of the EOF1 demonstrates the dominant vertical mode structure (Figs. 10a, 11a, 12a) and its temporal model variability (Figs 10b, 11b, and 12b) relative to the mean flow. Temporal mode variability displays the time-varying element of the vertical mode structure, which enables us to calculate the velocity variability relative to the mean flow (by multiplying the vertical mode structures (Figs. 10a, 11a, 12a) and the temporal variability (Figs 10b, 11b, and 12b).

Dominant dynamical processes are obtained by analyzing the velocity variability relative to the mean flow and by finding out dominant frequencies in the eigenperiod of mode-1. In this case, one can apply Fourier transform of these time series (Figs. 10b, 11b, and 12b) by assuming that the time varying signal is linear and periodic. In here, we use time series analysis technique (EEMD) which is using Hilbert transform, that suitable not only for linear periodic time series, but also suitable for nonlinear and nonperiodic signals. For detailed technique, please refer to Huang et al., 1998. Our results are shown in Figs. 10d-h, 11d-h, and 12d-h.

We do hope this could clarify it. Thank you.

2. The mesoscale eddies are very active in the study area, I think it should be important for the zonal current variations at intraseasonal to semi-annual time scale. Can the EOF analysis able to resolve the contribution of eddies? I strongly encourage the authors to discussion focus on this point of view in detail.

We would like to thank the reviewer for the insightful review of our manuscript. According to previous studies (e.g., Chelton et al., 2007; Yang et al., 2015), lifetime of mesoscale eddies in the SETIO is relatively short, approximately 50 days for cyclonic eddy and 46 days for anticyclonic eddy. In our present study, the time interval of the HYCOM data that used for the analysis is monthly mean. Hence, the highest (Nyquist) frequency resolved by the sample series is 0.5 cycles per month (periods ≥ 2 months). In here, we only focused on the analysis of the EOF1 of zonal current on timescale ≥ 2 months. Therefore, in this present study, the contribution of eddies could not be resolved yet.

Thank you for your insightful and encouraging suggestions. We really appreciate the suggestions. We would investigate this important topic concerning the contribution of eddies in the next study.

3. The relationship between IOD and zonal currents lies in two aspects: 1) the IOD SST anomaly induces surface wind anomalies, together with SSH gradient, resulting in zonal currents anomalies; and 2) Oceanic long waves dynamics (i.e., the Kelvin and Rossby waves) propagation may induce zonal currents anomalies and SSH anomalies in the southeastern tropical Indian Ocean, thereby resulting in IOD events. In addition, since the IOD tend to co-occur with ENSO, partial correlation is recommended to use, rather than correlation analysis.

We thank the referee for valuable recommendations. In the revised manuscript, we have calculated partial correlation (Table 7) and added additional descriptions as well (Lines 474-484), as follows:

*In addition to the lagged correlation analysis (Table 4), partial correlation analysis was also conducted since the IOD tend to co-occur with ENSO. Table 7 shows the partial correlation coefficients between zonal currents at 30 m on interannual timescale and the ONI and DMI. As for the ONI, the currents revealed significant positive correlations at $C_{EJ}$ during all monsoon seasons. This positive correlation suggests that El Niño (La Niña) events caused an eastward (westward) anomaly of currents at this point. Meanwhile, the partial correlation between the currents and the DMI showed significant negative correlation at $B_{SM}$, in which it occurred only during the SE monsoon (JJA), as shown in Table 7. This negative correlation indicates that an eastward (westward) anomaly of the currents was induced by negative (positive) IOD. The results of the partial correlation analysis confirm and complement the previous findings in Table 4 that ENSO mainly contributed to the zonal current variability at $C_{EJ}$ in DJF, MAM, JJA, and SON, whereas the IOD had a significant influence on the variability of current at $B_{SM}$ and only in JJA. In this present study, however, what are the causes of the influence of IOD on the current variability at $B_{SM}$ only in JJA are still unresolved. Further research is necessary to explain the dynamical links of this matter.*

Table 7. Partial correlation coefficients between zonal currents at 30 m on interannual timescale and each ONI and DMI. Only values above 95% confidence level are shown.

| Points | ONI – *U* (no DMI) | | | | DMI – *U* (no ONI) | | | |
|---|---|---|---|---|---|---|---|---|
| | DJF | MAM | JJA | SON | DJF | MAM | JJA | SON |
| $A_{WJ}$ | - | - | - | - | - | - | - | - |
| $B_{SM}$ | - | - | - | - | - | - | -0.76 | - |
| $C_{EJ}$ | 0.46 | 0.28 | 0.47 | 0.43 | - | - | - | - |
| The 95% significance level | ±0.19 | ±0.25 | ±0.26 | ±0.23 | ±0.63 | ±0.49 | ±0.35 | ±0.41 |

4. It is interesting that the zonal currents are correlated with remote winds in both equatorial Indian and Pacific oceans, but the explanation for the dynamical links are weak. Why the zonal currents lag the Pacific winds by 4 months? What is the pathways for the Pacific signal?

We would like to thank the reviewer for the thoughtful and insightful comments towards improving our manuscript. In the revised manuscript, we have provided additional explanation concerning this matter (Lines 525-539), as follows:

*Previous study conducted by Wijffels and Meyers (2004) shows that the variability in the ITF region associated with Kelvin and Rossby waves originating in the Indian and Pacific Oceans, respectively. They have revealed the pathways for equatorial Pacific wind energy traveling down the Papuan/Australian shelf break and radiating westward-propagating Rossby Waves into the Banda Sea and southeast Indian Ocean (their Fig. 20). Hence, there is a contribution of the westward-propagating Rossby Waves to the ITF variability inside the Indonesian seas or at the ITF exit regions (Ombai and Lombok Straits, and Timor Passage), which lead into the SETIO as well as the western coast of Sumatra and the southern coast of Java. Our simulation (Fig. 2) clearly shows that ITF flowing from the exit passages of Indonesian seas (Lombok, Ombai, and Timor passages) feeds into the SETIO region. Moreover, Wijffels and Meyers (2004) have computed the remotely driven Pacific Rossby wave speeds as a function of latitude. The phase speeds have been compared with the theoretical Rossby wave speeds based on atlas of Chelton et al. (1998). In this study, we have estimated the travel time of the westward-propagating Rossby waves excited by the wind anomalies in the central and western Pacific to the SETIO, especially at point $C_{EJ}$, based on the pathways for the Pacific signals introduced by Wijffels and Meyers (2004). In general, it was found that the equatorial Pacific signals around 130° W took approximately 3.01 months to the $C_{EJ}$ based on the mean phase speed of about 0.2 cm s$^{-1}$ taken from Wijffels and Meyers (2004). This travel time estimation was within the range of the 4-month lags between the flows at $C_{EJ}$ and the Pacific winds derived from the lagged correlation analysis in Fig. 17.*

**Specific comments:**
1. In section 2, Line 100-105, it should better be 'Data and Methods', instead of 'Material and Methods'
Thank you very much for your correction. In the revised manuscript, we have changed "Material and Methods" to "Data and Methods" (Line 127).

2. For the validation of the HYCOM data, it is better to calculate the root mean square errors as well.

We thank the referee for valuable suggestions. In the revised manuscript, we have calculated the root mean square errors as well (Figure 1 and Lines 140-143).

3. I think it is necessary to introduce the time interval of the HYCOM data that used for the analysis, is it daily or monthly mean? If it is daily averaged, it may hard to resolve intraseasonal signal.

Thank you very much for your comments and suggestions. The time interval of the HYCOM data that used for the analysis is monthly mean. Based on the Nyquist sampling theorem, the highest frequency resolved by the sample series is 0.5 cycles per month. Therefore, intraseasonal signals with a minimum period of about 2 months can still be resolved. In the revised manuscript, we have introduced the time interval of the HYCOM data that used for the analysis (Line 129).

4. Line 134, since the EOF has been introduced in Introduction, so it does not necessary to repeat the full name 'Empirical Orthogonal Function' here.

Thank you for the reviewer's correction. The repetition of the full name 'Empirical Orthogonal Function' has been deleted (Line 153).

5. It should also introduce the method for calculating the power spectrum.

Thank you for your suggestion. In the revised manuscript, we have introduced the method for calculating the power spectrum (Lines 158 - 163), as follows:

*Furthermore, a power spectral analysis (Emery and Thomson, 2001) was applied to the EEMD results to identify dominant periods of the zonal current variability in the study area. The power spectral analysis is computed from a measured time series by cutting the time series into several segments and applying Fourier analysis to these segments. The contribution from individual Fourier harmonics was subsequently summed to derive total energy of time series. In addition, 95% confidence red noise level in power spectrum, specified to acquire accurate confidence thresholds for true periodic signatures, was calculated based on number of degrees of freedom in each frequency band (Mann and Lees, 1996).*

---

## Author Response (AR3)

**Author Technical Corrections in Response to the Topic Editor Report "Simulated Zonal Current Characteristics in the Southeastern Tropical Indian Ocean (SETIO)" - (OS-2020-91) by Nining Sari Ningsih et al.**

We thank the Topical Editor, Dr. Viviane Menezes, for guidance through this process. The final technical remarks are shown here in black text, whereas our response is in blue text.

Topic Editor Decision: Publish subject to technical corrections (02 Jul 2021) by Viviane Menezes

Comments to the Author:

Dear Dr. Ningsih,

We have received the two reviews of your MS entitled "Simulated Zonal Current Characteristics in the Southeastern Tropical Indian Ocean (SETIO)." Both reviewers agree that the new version is suitable for publication in Ocean Sciences, and all their concerns have been addressed, although one of them found the abstract too long. I have taken a quick look at the revised MS and agree with the reviewers-- the MS is much improved, but the abstract is excessive length. Generally, abstracts have between 150-300 words, and yours have 527. Also, there are too many acronyms.

Please take a look at the recommendation from the OSD website: "Abstract: the abstract should be intelligible to the general reader without reference to the text. After a brief introduction of the topic, the summary recapitulates the key points of the article and mentions possible directions for prospective research. Reference citations should not be included in this section unless urgently required, and abbreviations should not be included without explanations. An abstract should be short, clear, concise, and written in English with correct spelling and good sentence structure."

Thus, I am recommending the MS for publication subject to technical corrections.

Thank you very much. I am very glad to hear that the MS has been accepted for publication subject to technical corrections. On behalf of all authors, we really appreciate all your help and guidance with the manuscript revision. In addition, we thank the reviewers for the time and effort that they invested into the review of our manuscript, and for their helpful comments, corrections, and suggestions, which helped make our manuscript stronger.

The abstract must be shortened.

Thank you for the thoughtful suggestions and comments towards improving our manuscript. We have shortened and refined the abstract. In the revised version, it has 300 words (L# 9–26), as follows:

*Detailed ocean currents in the Southeastern Tropical Indian Ocean adjacent to southern Sumatra-Java coasts have not been fully explained because of limited observations. In this study, zonal current characteristics in the region have been studied using simulation results of a 1/8º global HYbrid Coordinate Ocean Model from 1950 to 2013. The simulated zonal currents across three meridional sections were then investigated using an empirical orthogonal function (EOF), where the first three modes account for 75–98% of the total variance. The first temporal mode of EOF is then investigated using ensemble empirical mode decomposition (EEMD) to distinguish the signals.*
*This study has revealed distinctive features of currents in the South Java Current (SJC) region, the Indonesian Throughflow (ITF)/South Equatorial Current (SEC) region, and the transition zone between these regions. The vertical structures of zonal currents in south Java and offshore Sumatra are characterized by a one-layer flow. Conversely, a two-layer flow is observed in the nearshore and transition regions of Sumatra. Current variation in the SJC region has peak energies, which are sequentially dominated by semiannual, intraseasonal, and annual timescales. Meanwhile, the transition*

*zone is characterized by semiannual and intraseasonal periods with pronounced interannual variations. In contrast, interannual variability associated with ENSO and IOD modulates the prominent intraseasonal variability of current in the ITF/SEC region. ENSO has the strongest influence at the outflow ITF, while IOD's strongest influence at southwest Sumatra, with the ENSO (IOD) leading the current by four months (one month). Moreover, the contributions (largest to smallest) of each EEMD mode at the nearshore of Java and offshore Sumatra are intraseasonal, semiannual, annual, interannual, and long-term fluctuations. The contribution of long-term variation (19.2%) at far offshore eastern Indian Ocean is larger than the interannual (16.3%) and annual (14.7%) variations. Future studies should be conducted to investigate this long-term variation.*

Also, in the Introduction, two phrases in sequence (L#58-L#60) start with "Meanwhile" which makes the text a bit hard to understand. The authors may want to fix this issue as well. L#58-60: Meanwhile, 60-day variations are the dominant feature in the SJUC, which are forced by intraseasonal atmospheric variability associated with the eastward movement of the Madden-Julian Oscillation (MJO) over the eastern equatorial Indian Ocean. Meanwhile, seasonal variabilities of SJC and SJUC that exist along the coasts of western Sumatra and southern Java have been investigated based on observation data (e.g., Sprintall et al., 1999; 2010; Qu and Meyers, 2005).

We would like to thank you for the insightful and careful review of our manuscript. We have fixed this issue (L#58–60 in the previous MS; L#47–50 in the revised MS), as follows:

*Meanwhile, 60-day variations are the dominant feature in the SJUC, which are forced by intraseasonal atmospheric variability associated with the eastward movement of the Madden-Julian Oscillation (MJO) over the eastern equatorial Indian Ocean.*
*On a seasonal timescale, variabilities of SJC and SJUC that exist along the coasts of western Sumatra and southern Java have been investigated based on observation data (e.g., Sprintall et al., 1999; 2010; Qu and Meyers, 2005).*

I look forward to receiving the revised manuscript soon.

Best regards,
Viviane Menezes

We thank the Topical Editor and the OS editorial support team for wise advice and support during this process.